

# Separability and entanglement of resonating valence-bond states

Gilles Parez[1*], Clément Berthiere[1,2†] and William Witczak-Krempa[1,2,3‡]

**1** Centre de Recherches Mathématiques, Université de Montréal,
Montréal, QC H3C 3J7, Canada
**2** Département de Physique, Université de Montréal, Montréal, QC H3C 3J7, Canada
**3** Institut Courtois, Université de Montréal, Montréal, QC H2V 0B3, Canada

★ gilles.parez@umontreal.ca , † clement.berthiere@umontreal.ca ,
‡ w.witczak-krempa@umontreal.ca

## Abstract

We investigate separability and entanglement of Rokhsar-Kivelson (RK) states and resonating valence-bond (RVB) states. These states play a prominent role in condensed matter physics, as they can describe quantum spin liquids and quantum critical states of matter, depending on their underlying lattices. For dimer RK states on arbitrary tileable graphs, we prove the exact separability of the reduced density matrix of $k$ disconnected subsystems, implying the absence of bipartite and multipartite entanglement between the subsystems. For more general RK states with local constraints, we argue separability in the thermodynamic limit, and show that any local RK state has zero logarithmic negativity, even if the density matrix is not exactly separable. In the case of adjacent subsystems, we find an exact expression for the logarithmic negativity in terms of partition functions of the underlying statistical model. For RVB states, we show separability for disconnected subsystems up to exponentially small terms in the distance $d$ between the subsystems, and that the logarithmic negativity is exponentially suppressed with $d$. We argue that separability does hold in the scaling limit, even for arbitrarily small ratio $d/L$, where $L$ is the characteristic size of the subsystems. Our results hold for arbitrary lattices, and encompass a large class of RK and RVB states, which include certain gapped quantum spin liquids and gapless quantum critical systems.



# 1 Introduction

Arguably the most fascinating phenomenon of quantum mechanics, entanglement has confounded many a physicist since Einstein, Podolsky, Rosen [1] and Schrödinger [2]. Once mainly a subject of philosophical debates, entanglement now constitutes a central notion in the modern field of quantum information [3], where it is recognized as a resource, enabling tasks such as quantum cryptography [4] or quantum teleportation [5].

More recently, entanglement has been shown to play a prominent role in quantum many-body systems [6–8]. In particular, groundstate entanglement of many-body Hamiltonians is related to critical properties [9–11] and topological order [12, 13]. The detection and quantification of entanglement is a fundamental issue, and despite a considerable amount of work [14–16], it still remains extremely challenging to determine whether a given quantum state is entangled or separable, and no general solution to the *separability problem* is known as of yet.

Let $\rho_{A_1 \cup A_2}$ act on the Hilbert space $\mathcal{H} = \mathcal{H}_{A_1} \otimes \mathcal{H}_{A_2}$. A state $\rho_{A_1 \cup A_2}$ is called separable [17,18] if it can be written as a finite convex combination of pure product states $\rho_{A_1}^{(i)} \otimes \rho_{A_2}^{(j)}$, i.e.

$$\rho_{A_1 \cup A_2} = \sum_{i,j} p_{ij} \rho_{A_1}^{(i)} \otimes \rho_{A_2}^{(j)}, \tag{1}$$

where the probabilities $p_{ij}$ sum to one. This definition of separability usually requires $\rho_{A_k}^{(\ell)}$ to

be projectors on normalized pure states. However, since any mixed state can be written as a convex sum of pure states, it suffices that $\rho_{A_k}^{(\ell)}$ be Hermitian positive semidefinite operators.

There exist several criteria that imply that a state is entangled or not. The quintessential example is the entanglement entropy for bipartite pure states; if it vanishes, then the state is separable. For mixed states, detecting entanglement reveals to be more complicated. A simple computable measure of entanglement for mixed states is the logarithmic negativity [19–21], which is based on the positive partial transpose (PPT) criterion [22,23], and defined as

$$\mathcal{E}(A_1 : A_2) = \log \mathrm{Tr} \left| \rho_{A_1 \cup A_2}^{T_1} \right|, \tag{2}$$

where $\mathrm{Tr}|O| \equiv \mathrm{Tr}\sqrt{O^\dagger O}$ is the trace-norm of $O$, and $\rho_{A_1 \cup A_2}^{T_1}$ is the partial transpose of $\rho_{A_1 \cup A_2}$ with respect to the degrees of freedom of $A_1$. A vanishing logarithmic negativity provides, in general, only a necessary but not sufficient condition for separability, i.e. there exist entangled states that remain positive under partial transposition (PPT states) [18]. Such states have the interesting property that their entanglement cannot be distilled.

The definition of separability and entanglement is more complex in the multipartite scenario than in the bipartite case, see [15,16] and references therein. Full separability, a direct extension of bipartite separability, exists along with various forms of partial separability. For instance, a state which is separable for each possible bipartition is not necessarily fully separable. The structure of entanglement is much richer when more than two parties are involved. In particular, several inequivalent classes of entanglement can be identified. To fully characterize the entanglement structure of a system, it is thus crucial to investigate its multipartite entanglement and separability properties. Recently, there has been a burst of theoretical activities aiming at better understanding multipartite entanglement in quantum many-body systems, both in [24–31] and out of equilibrium [32–34].

In this paper, we investigate entanglement and separability of Rokhsar-Kivelson (RK) states and resonating valence-bond (RVB) states. Introduced by Anderson [35,36] as trial groundstates for the anti-ferromagnetic spin-1/2 Heisenberg chain on the triangular lattice, such RVB states are celebrated instances of quantum spin liquid where pairs of electrons form singlet (valence) bonds, a superposition of which yields a liquidlike, non-Néel groundstate. Quantum spin liquids are phases of matter with no long-range order which exhibit exotic features arising from their topological nature [37,38], such as fractional excitations [39], spin-charge separation [38], protected groundstate degeneracy [40–42] and relation to gauge theory [43–46]. A unifying and essential property of spin liquids is long-range entanglement, which implies that the wavefunction cannot be continuously deformed into a product state. Since entanglement plays such an important role in the definition and properties of quantum spin liquids, it is natural to investigate their expected representatives through that lens (see, e.g., [8,12,13,47–51]).

Quantum dimer models are paradigmatic examples of strongly-correlated systems subject to hard local constraints. They were originally introduced on the square lattice by Rokhsar and Kivelson [40,52] to describe the low-energy physics of short-range RVB states; here a valence bond is represented by a dimer linking the two electrons which form it. Crucially, quantum dimer models exhibit an "RK point" where the wavefunction is an equal-weight superposition of all dimer coverings, which is the characteristic RVB form. The dimer RK wavefunction is known to be a critical liquid state on the square lattice [53,54], whereas a gapped $\mathbb{Z}_2$ liquid state is realized on triangular and kagome (frustrated) lattices [55–57]. Dimer and RVB states have also been investigated on three-dimensional lattices [58]. Similarly as in two dimensions, they may describe critical or gapped phases, depending on whether the underlying lattice is bipartite or not. Quantum dimer models thus come in many different flavors. Their study have unearthed a wealth of phenomena, such as rich phase diagrams [59–64], mapping to height models [65,66], gauge theory [46,57,67], and more.

The construction of RK states is not limited to lattice models, nor are the wavefunctions required to be equal-weight superpositions of all configurations [66, 68, 69]; one can, e.g., construct an RK state from the Boltzmann weights of their favorite statistical model. Some entanglement properties of RK states have been studied in [70–76]. Recently, continuum RK states for which the underlying models are local quantum field theories (QFTs) have been shown to be separable for two disconnected regions [77] (see also [78]), which can be traced back to the locality of the theory. In particular, taking the local QFT to be the free scalar field describes the continuum limit of the dimer RK and RVB wavefunctions on the square lattice [51, 66, 68, 79]. We note that separability implies a vanishing logarithmic negativity, and mention that the logarithmic negativity for disjoint subsystems vanishes for other systems as well, such as the toric code [80, 81], the AKLT model [82, 83], Motzkin and Fredkin spin chains [84,85], and Chern-Simons theories [86,87]. Inspired by these results, one may wonder whether separability and vanishing logarithmic negativity hold for dimer and more general RK states on arbitrary graphs, as well as for RVB states. The goal of this work is therefore to address this important issue.

This paper is organized as follows. We start in Sec. 2 with RK states. We study the separability of the reduced density matrix of two disconnected subsystems, for dimer RK states and more general RK states with local constraints, on arbitrary graphs. We give general expressions for the logarithmic negativity of such states at the end of the section, both for disconnected and adjacent subsystems. In Sec. 3, we study the separability of RVB states on arbitrary graphs. We discuss their logarithmic negativity as well as relevant higher-spin generalizations at the end of the section. Finally, we investigate multipartite separability of RK and RVB states in Sec. 4. We conclude in Sec. 5 with a summary of our main results, and give an outlook on future study.

## 2 Rokhsar-Kivelson states

In this section, we review the definition of RK states and investigate their separability. These are quantum states whose Hilbert space is spanned by the configurations of an underlying statistical model.

### 2.1 Definition

Consider a statistical model on an arbitrary graph, with allowed configurations $c \in \Omega$, "energy" functional $E(c)$, Boltzmann weights $\mathrm{e}^{-E(c)}$ and partition function $\mathcal{Z}$. For each configuration $c$, we assign a quantum state $|c\rangle$ and impose $\langle c|c'\rangle = \delta_{c,c'}$. The corresponding normalized RK state is

$$|\psi\rangle = \frac{1}{\sqrt{\mathcal{Z}}} \sum_{c \in \Omega} \mathrm{e}^{-\frac{1}{2}E(c)} |c\rangle, \qquad \mathcal{Z} = \sum_{c \in \Omega} \mathrm{e}^{-E(c)}. \tag{3}$$

Different underlying statistical models yield different RK states. We shall focus on RK states built from models whose degrees of freedom reside on the edges of the graph, with a local energy functional. Moreover, we assume that the models satisfy local constraints, where the state of all but one edge connected to a common vertex fixes the state of the remaining edge. Such models include vertex models with generalized ice-rule and dimer models.

### 2.2 Tripartition and disconnected subsystems

Let the underlying statistical model be defined on a graph which consists of three subregions, $A_1$, $A_2$ and $B$. In this setting, a subregion is a set of edges of the graph. Two edges are said to be adjacent if they are connected to a common vertex. We assume that $A_1$ and $A_2$ are

disconnected, namely edges in $A_1$ and $A_2$ are never adjacent. By convention, the boundary between $A_1, A_2$ and $B$ consists of the edges in $A_1, A_2$ that are adjacent to edges in $B$. We denote the configurations on these boundaries by $i$ and $j$, respectively. In contrast, the bulk configurations of $A_1, A_2$ (and $B$) do not include the boundary edges. We illustrate such a tripartition in Fig. 1 for the dimer model on the square lattice.

The state corresponding to a configuration $c$ can be decomposed as

$$|c\rangle = |a_1, i\rangle \otimes |b\rangle \otimes |a_2, j\rangle. \tag{4}$$

Here, $a_1, a_2$ are bulk configurations of $A_1, A_2$, while $b$ is the configuration of $B$, and $i, j$ are the boundary configurations. We have $\langle a_k, \ell | a_k', \ell' \rangle = \delta_{a_k, a_k'} \delta_{\ell, \ell'}$ with $k = 1, 2$, $\ell = i, j$, as well as $\langle b | b' \rangle = \delta_{b, b'}$. We denote by $\Omega_{A_k}^\ell$ the set of all bulk configurations of $A_k$ that are compatible with the boundary configuration $\ell$. Similarly, $\Omega_B^{ij}$ is the set of all configurations of $B$ compatible with both boundary configurations. Moreover, because the energy functional $E(c)$ is local, we may express it as

$$E(c) = E(a_1, i) + E(b, i, j) + E(a_2, j), \tag{5}$$

where $E(a_k, \ell)$ encodes the interaction in the bulk of subsystem $A_k$, as well as interactions between bulk and boundary degrees of freedom. It is similar for $E(b, i, j)$, except $B$ has degrees of freedom adjacent to both boundaries $i$ and $j$.

With these conventions, the RK wavefunction (3) reads

$$|\psi\rangle = \sum_{i,j} \left( \frac{\mathcal{Z}_{A_1}^i \, \mathcal{Z}_{A_2}^j \, \mathcal{Z}_B^{ij}}{\mathcal{Z}} \right)^{1/2} |\psi_{A_1}^i\rangle \otimes |\psi_B^{ij}\rangle \otimes |\psi_{A_2}^j\rangle, \tag{6a}$$

with subsystem RK states

$$|\psi_{A_k}^\ell\rangle = \frac{1}{\sqrt{\mathcal{Z}_{A_k}^\ell}} \sum_{a_k \in \Omega_{A_k}^\ell} e^{-\frac{1}{2} E(a_k, \ell)} |a_k, \ell\rangle,$$

$$|\psi_B^{ij}\rangle = \frac{1}{\sqrt{\mathcal{Z}_B^{ij}}} \sum_{b \in \Omega_B^{ij}} e^{-\frac{1}{2} E(b, i, j)} |b\rangle, \tag{6b}$$

and the normalizations

$$\mathcal{Z}_{A_k}^\ell = \sum_{a_k \in \Omega_{A_k}^\ell} e^{-E(a_k, \ell)}, \qquad \mathcal{Z}_B^{ij} = \sum_{b \in \Omega_B^{ij}} e^{-E(b, i, j)}. \tag{6c}$$

## 2.3 Reduced density matrix

In this section, we compute the RK reduced density matrix $\rho_{A_1 \cup A_2} = \mathrm{Tr}_B(|\psi\rangle\langle\psi|)$ of the subsystem $A_1 \cup A_2$. The calculation depends on the underlying statistical model, the lattice and the shapes of the subsystems.

### 2.3.1 Arbitrary graphs

Let us consider an RK state defined on an arbitrary graph. We only impose that the two regions $A_1$ and $A_2$ are disconnected. In general, there might be vertices connected to edges in $B$ and to more than one edge in $A_1$ or $A_2$. This is for example the case for the square lattice in the case where the boundaries have concave angles, or the triangular lattice, see Fig. 2 below. Hence, there may be different boundary configurations compatible with the same configurations in $B$.

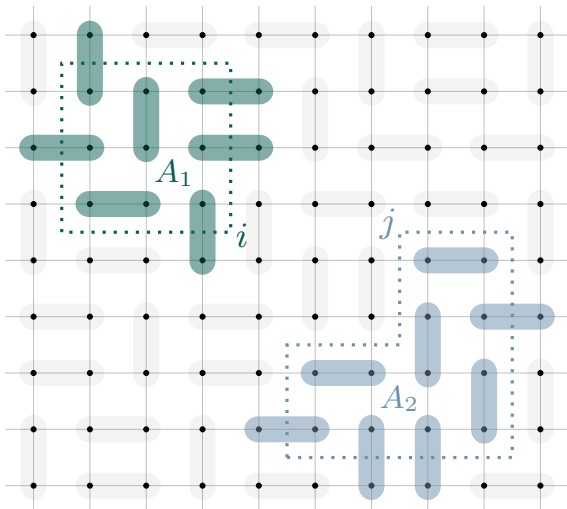

Figure 1: Illustration of a tripartite geometry for a specific configuration of the dimer model on the square lattice. Regions $A_1$ and $A_2$ are tiled with green and blue dimers, respectively, and consist of the edges encircled or crossed by the dotted lines; region $B$ is tiled with gray dimers. The boundary dimers are those that cross the boundaries (dotted lines) of the subsystems. Indices $i$ and $j$ correspond to the boundary configurations between $B$ and $A_1$ or $A_2$, respectively.

To proceed, we introduce the notation $i \sim i'$ for boundary configurations $i, i'$ that are compatible with the same configurations in $B$. By definition, we also have $i \sim i$, namely we do not impose that $i \neq i'$. This translates to

$$\Omega_B^{ij} = \Omega_B^{i'j'}, \qquad i \sim i', \ j \sim j', \tag{7}$$

and we have the orthogonality relation

$$\langle \psi_B^{ij} | \psi_B^{i'j'} \rangle = \delta_{i \sim i'} \delta_{j \sim j'} \frac{\mathcal{Z}_B^{ij,i'j'}}{\sqrt{\mathcal{Z}_B^{ij} \mathcal{Z}_B^{i'j'}}}, \tag{8}$$

where $\delta_{i \sim i'} = 1$ if $i \sim i'$, and vanishes otherwise. Moreover, we introduced

$$\mathcal{Z}_B^{ij,i'j'} = \sum_{b \in \Omega_B^{ij}} e^{-\frac{1}{2}(E(b,i,j) + E(b,i',j'))}. \tag{9}$$

The reduced density matrix reads

$$\rho_{A_1 \cup A_2} = \sum_{i,j} \sum_{i' \sim i} \sum_{j' \sim j} P_{ij,i'j'} |\psi_{A_1}^i\rangle \langle \psi_{A_1}^{i'}| \otimes |\psi_{A_2}^j\rangle \langle \psi_{A_2}^{j'}|, \tag{10a}$$

with

$$P_{ij,i'j'} = \frac{(\mathcal{Z}_{A_1}^i \mathcal{Z}_{A_1}^{i'} \mathcal{Z}_{A_2}^j \mathcal{Z}_{A_2}^{j'})^{1/2} \mathcal{Z}_B^{ij,i'j'}}{\mathcal{Z}}. \tag{10b}$$

### 2.3.2 Square lattice and no concave angles

Let us assume that the graph is the two-dimensional square lattice, and the subsystems $A_1$ and $A_2$ do not have any concave angles (they can be rectangles, strips, cylinders, etc). In that case, the calculation of the reduced density matrix simplifies greatly.

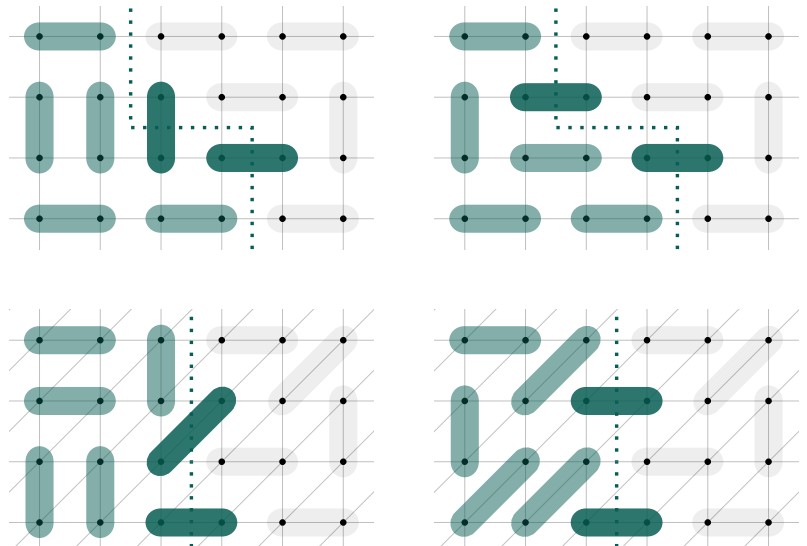

Figure 2: Illustration of two different configurations of the dimer model for a region with a concave angle (top) and on the triangular lattice (bottom). In both cases, the two configurations have different boundary configurations (highlighted darker green dimers), but are both compatible with the same configuration of dimers outside the green region.

If a configuration $b$ of $B$ is compatible with a boundary configuration $(i, j)$, then the local constraints imply that $b$ is incompatible with all other possible choices $(i', j') \neq (i, j)$. In other words,

$$\Omega_B^{ij} \cap \Omega_B^{i'j'} = \emptyset, \qquad (i, j) \neq (i', j'), \tag{11}$$

and the relation (8) becomes $\langle \psi_B^{i'j'} | \psi_B^{ij} \rangle = \delta_{i,i'} \delta_{j,j'}$.

The density matrix $\rho = |\psi\rangle \langle \psi|$ is a double sum over the pairs of indices $(i, j)$ and $(i', j')$ that involve projectors of the form $|\psi_B^{ij}\rangle \langle \psi_B^{i'j'}|$. Using the orthogonality of the RK wavefunctions for $B$, we obtain

$$\rho_{A_1 \cup A_2} = \sum_{i,j} \frac{\mathcal{Z}_{A_1}^i \mathcal{Z}_{A_2}^j \mathcal{Z}_B^{ij}}{\mathcal{Z}} |\psi_{A_1}^i\rangle \langle \psi_{A_1}^i| \otimes |\psi_{A_2}^j\rangle \langle \psi_{A_2}^j|. \tag{12}$$

We note that this is a simplification of (10), because in this case $\delta_{i \sim i'} = \delta_{i,i'}$.

The reduced density matrix (12) can be cast in the form

$$\rho_{A_1 \cup A_2} = \sum_{i,j} p_{ij} \rho_{A_1}^{(i)} \otimes \rho_{A_2}^{(j)}, \tag{13a}$$

with

$$p_{ij} = \frac{\mathcal{Z}_{A_1}^i \mathcal{Z}_{A_2}^j \mathcal{Z}_B^{ij}}{\mathcal{Z}}, \qquad \rho_{A_k}^{(\ell)} = |\psi_{A_k}^\ell\rangle \langle \psi_{A_k}^\ell|. \tag{13b}$$

Here, $\rho_{A_k}^{(\ell)}$ are pure states, and hence the reduced density matrix $\rho_{A_1 \cup A_2}$ is separable in the sense of (1).

## 2.4 Separability for disconnected subsystems

For disjoint $A_1$ and $A_2$ with no concave angles on the square lattice, we showed with (13a) that the reduced density matrix for any RK state with local constraints is separable. For the

more general situation of disjoint subsystems with concave angles and/or a model defined on an arbitrary lattice, the reduced density matrix given in (10) is not trivially separable. We investigate the separability of the reduced density matrix in this case.

### 2.4.1 Dimer states

We first focus on RK states whose underlying statistical model is the dimer model. An allowed configuration of dimers on a graph, or tiling, is such that each vertex is covered by exactly one dimer, and allowed configurations have the same Boltzmann weight. Dimer states are thus particular types of RK states, where $E(c) = 0$ for allowed dimer configurations, and $E(c) = \infty$ for forbidden ones.

Since all allowed configurations have the same Boltzmann weight, and using (7), we have

$$\mathcal{Z}_B^{ij,i'j'} = \mathcal{Z}_B^{ij} = \mathcal{Z}_B^{i'j'} = \mathcal{Z}_B^{ij'} = \mathcal{Z}_B^{i'j}, \qquad i \sim i', \; j \sim j', \tag{14}$$

such that

$$P_{ij,i'j'} = \frac{(\mathcal{Z}_{A_1}^i \, \mathcal{Z}_{A_1}^{i'} \, \mathcal{Z}_{A_2}^j \, \mathcal{Z}_{A_2}^{j'} \, \mathcal{Z}_B^{ij} \, \mathcal{Z}_B^{i'j'})^{1/2}}{\mathcal{Z}} \,. \tag{15}$$

From $P_{ij,i'j'}$ in (15) and the symmetry of $\mathcal{Z}_B^{ij}$ given in (14), the reduced density matrix (10) is symmetric in $i \leftrightarrow i'$ and $j \leftrightarrow j'$. In particular, we may rewrite it as

$$\rho_{A_1 \cup A_2} = \sum_{i,j} \frac{\mathcal{Z}_{A_1}^i \, \mathcal{Z}_{A_2}^j \, \mathcal{Z}_B^{ij}}{\mathcal{Z}} \rho_{A_1}^{(i)} \otimes \rho_{A_2}^{(j)}, \tag{16a}$$

where we explicitly symmetrized the density matrices,

$$\rho_{A_k}^{(\ell)} = \frac{1}{2} \sum_{\ell' \sim \ell} \sqrt{\frac{\mathcal{Z}_{A_k}^{\ell'}}{\mathcal{Z}_{A_k}^{\ell}}} \left( |\psi_{A_k}^{\ell}\rangle\langle\psi_{A_k}^{\ell'}| + |\psi_{A_k}^{\ell'}\rangle\langle\psi_{A_k}^{\ell}| \right). \tag{16b}$$

The reduced density matrix corresponding to two disjoint regions is hence separable. As a consistency check, we note that (16b) reduces to (13b) for regions with no concave angles on the square lattice.

We thus conclude that for the dimer RK states, two disconnected regions are *not* entangled. This is in accordance with the result of [77] where it was shown that continuum RK states are separable if the subsystem consists of two disjoint regions. However, we emphasize that here, we prove exact separability on the lattice, without taking any thermodynamic/continuum limit.

### 2.4.2 Rokhsar-Kivelson states with local constraints

Taking a generic underlying statistical model (still satisfying local constraints), we have $\Omega_B^{ij} = \Omega_B^{i'j'}$ for $i \sim i'$ and $j \sim j'$. However, in general we cannot absorb the sums over $i'$ and $j'$ separately to define reduced density matrices for $A_1$ and $A_2$, as in (16). This issue arises because of the term $\mathcal{Z}_B^{ij,i'j'}$ in $P_{ij,i'j'}$, see (10). We can however argue that the reduced density matrix $\rho_{A_1 \cup A_2}$ is nearly separable in the thermodynamic limit where the volume of each subsystem $A_1, A_2, B$ becomes large, whereas their ratio is kept constant. We stress that the following argument also holds in the limit where $B$ becomes large with $A_1, A_2$ finite.

Owing to the locality of the energy functional and the fact that $A_1$ and $A_2$ are disjoints, we may express $E(b, i, j)$ as

$$E(b, i, j) = E_{\text{bulk}}(b) + E_{\text{bd}}(b, i) + E_{\text{bd}}(b, j), \tag{17}$$

where $E_{\text{bulk}}(b)$ encodes the bulk energy of the configuration $b$, whereas $E_{\text{bd}}(b,i)$ is the energy arising from the interactions between $B$ and the boundary $i$.

In general, we can write

$$E(b,i,j) = E_{\text{bulk}}(b)(1 + \Delta_{ij}),\tag{18}$$

and we expect $|\Delta_{ij}| \ll 1$, because boundary energies are negligible compare to bulk energies in the thermodynamic limit. We thus approximate $\mathcal{Z}_B^{ij,i'j'}$ as

$$\mathcal{Z}_B^{ij,i'j'} \simeq \sum_{b \in \Omega_B^{ij}} e^{-E_{\text{bulk}}(b)} \equiv \mathcal{Z}_{B,\text{bulk}}^{ij}.\tag{19}$$

Since by definition $\mathcal{Z}_{B,\text{bulk}}^{ij} = \mathcal{Z}_{B,\text{bulk}}^{i'j} = \mathcal{Z}_{B,\text{bulk}}^{ij'} = \mathcal{Z}_{B,\text{bulk}}^{i'j'}$ for $i \sim i'$ and $j \sim j'$, the construction of the previous section holds, and the reduced density matrix takes the separable form of (16) where $\mathcal{Z}_B^{ij}$ is replaced by $\mathcal{Z}_{B,\text{bulk}}^{ij}$. Again, this is in agreement with the separability of continuum RK states for disconnected subsystems [77].

## 2.5 Logarithmic negativity

As alluded to in the introduction, the logarithmic negativity is given as the violation of the PPT criterion and serves as a measure of entanglement for mixed states. In its original definition (2), the logarithmic negativity requires the knowledge of the spectrum of $\rho_{A_1 \cup A_2}^{T_1}$, which is very difficult to obtain for quantum many-body systems. To circumvent this difficulty, a replica method was developed in [88,89], which relates the logarithmic negativity to the moments of $\rho_{A_1 \cup A_2}^{T_2}$, i.e.

$$\mathcal{E}(A_1 : A_2) = \lim_{n \to 1/2} \log \text{Tr}\left(\rho_{A_1 \cup A_2}^{T_1}\right)^{2n}.\tag{20}$$

For pure states, the logarithmic negativity reduces to the Rényi entropy of order $n = 1/2$, defined as

$$S_n(A_1) = \frac{1}{1-n} \log \text{Tr} \rho_{A_1}^n,\tag{21}$$

for the reduced density matrix $\rho_{A_1}$. We shall give general expressions for the logarithmic negativity of RK states.

### 2.5.1 Disjoint subsystems

The reduced density matrix $\rho_{A_1 \cup A_2}$ for disjoint subsystems is given in (10). Using (9) and (17), one may readily verify that $P_{ij,i'j'} = P_{i'j,ij'}$, hence $\rho_{A_1 \cup A_2} = \rho_{A_1 \cup A_2}^{T_1}$. This implies $\text{Tr}|\rho_{A_1 \cup A_2}^{T_1}| = \text{Tr} \rho_{A_1 \cup A_2} = 1$ and thus a vanishing logarithmic negativity,

$$\mathcal{E}(A_1 : A_2) = 0.\tag{22}$$

We conclude that any RK state with local constraints has zero negativity, even if the density matrix is not exactly separable. Note that for dimer states, the exact separability implies a vanishing negativity.

In the previous sections, we focused on RK states with local constraints. However, our arguments can be generalized to RK states for which the local constraints rule is removed. An example would be an RK state constructed from an underlying Ising model where the spins are defined on the vertices of the graph. There are no constraints in that case, hence all boundary configurations are compatible with all bulk configurations of $B$. Expression (10) remains valid, only the sum $\sum_{i' \sim i}$ becomes a sum over all possible configurations $i'$, irrespective of $i$, and similarly for $j, j'$. The locality of the energy functional still implies that (17) holds, such that we have $\rho_{A_1 \cup A_2}^{T_1} = \rho_{A_1 \cup A_2}$ and a vanishing negativity.

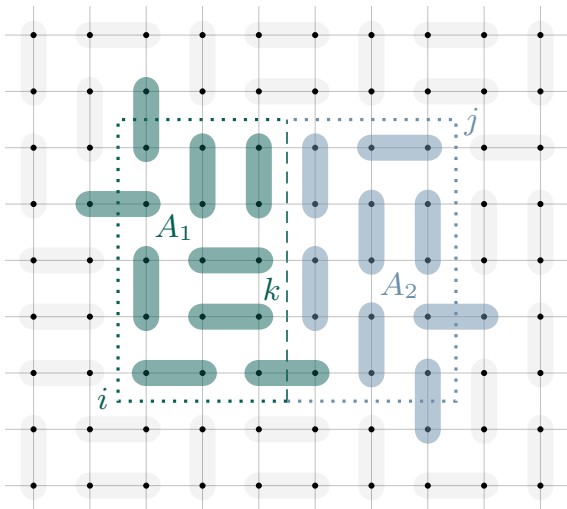

Figure 3: Illustration of a tripartite geometry where regions $A_1$ and $A_2$ are adjacent for a dimer state. The boundary dimers between $A_1$, $A_2$ and $B$ are those that cross the boundaries of the subsystems. Indices $i$ and $i$ (dotted lines) correspond to the boundary configurations of $A_1$ and $A_2$ with respect to $B$, respectively, while $k$ (dashed line) denotes the boundary configuration between $A_1$ and $A_2$.

### 2.5.2 Comment on the mutual information

Commonly used as a measure of entanglement and correlations between separate subsystems, the mutual information is defined as

$$I(A_1 : A_2) = S(A_1) + S(A_1) - S(A_1 \cup A_2),\tag{23}$$

where $S(A) = \lim_{n \to 1} S_n(A)$ is the celebrated entanglement entropy. With our results from the previous sections and that of [72], one can express the mutual information of RK states in terms of partition functions of the underlying model. In particular, it does not vanish identically for disconnected systems, contrarily to the logarithmic negativity. The mutual information has a well defined operational meaning [90] as the total amount of correlations, both quantum and classical, between two systems, whereas the logarithmic negativity is a genuine quantum entanglement measure [21]. Separability of RK states then implies that the mutual information results entirely from classical and quantum non-entangling correlations [91, 92].

### 2.5.3 Adjacent subsystems

For two adjacent subsystems $A_1$ and $A_2$, the corresponding reduced density matrix $\rho_{A_1 \cup A_2}$ is in general not separable. Below, we derive an explicit expression for the logarithmic negativity in terms of partition functions of the underlying statistical model, similarly as for the Rényi entropies, see [72].

As for the disjoint case, the boundary configurations between $B$ and $A_1$ and $A_2$ are denoted $i$ and $j$, respectively. By convention, the edges that connect $A_1$ and $A_2$ belong to $A_1$, and the corresponding boundary configurations are denoted $k$. We illustrate this geometry in Fig. 3 for the dimer state. Using similar conventions as in previous sections, the RK state (3) for two adjacent subsystems can be written as

$$|\psi\rangle = \sum_{i,j,k} \left( \frac{\mathcal{Z}_{A_1}^{ik} \mathcal{Z}_{A_2}^{jk} \mathcal{Z}_B^{ij}}{\mathcal{Z}} \right)^{1/2} |\psi_{A_1}^{ik}\rangle \otimes |\psi_{A_2}^{jk}\rangle \otimes |\psi_B^{ij}\rangle,\tag{24}$$

where the partition functions and RK states for $A_1$ and $A_2$ are defined as in (6), but now also depend on their common boundary configuration $k$. For convenience, we introduce the probabilities $p_{ijk}$ as

$$p_{ijk} = \frac{\mathcal{Z}_{A_1}^{ik} \mathcal{Z}_{A_2}^{jk} \mathcal{Z}_B^{ij}}{\mathcal{Z}} . \tag{25}$$

For simplicity, we consider RK states with local constraints on the square lattice and boundaries with no concave angles, as in Fig. 3. The following calculations can be generalized to arbitrary situations with the technical tools developed in Sec. 2.3.1. With these constraints, RK states for $B$ are orthogonal, and hence the reduced density matrix reads

$$\rho_{A_1 \cup A_2} = \sum_{i,j,k,\ell} (p_{ijk} p_{ij\ell})^{1/2} |\psi_{A_1}^{ik}\rangle \langle \psi_{A_1}^{i\ell}| \otimes |\psi_{A_2}^{jk}\rangle \langle \psi_{A_2}^{j\ell}| . \tag{26}$$

We now compute the logarithmic negativity using the replica definition (20). To proceed, the partial transposition of $\rho_{A_1 \cup A_2}$ with respect to $A_1$ reads

$$\rho_{A_1 \cup A_2}^{T_1} = \sum_{i,j,k,\ell} (p_{ijk} p_{ij\ell})^{1/2} |\psi_{A_1}^{i\ell}\rangle \langle \psi_{A_1}^{ik}| \otimes |\psi_{A_2}^{jk}\rangle \langle \psi_{A_2}^{j\ell}| . \tag{27}$$

Since there are no concave angles in the boundary between $A_1$ and $A_2$, their respective RK states are orthogonal, and we find

$$\mathrm{Tr}(\rho_{A_1 \cup A_2}^{T_1})^{2n} = \sum_{i,j,k,\ell} p_{ijk}^n p_{ij\ell}^n , \tag{28}$$

for integer values of $n$. The limit $n \to 1/2$ yields

$$\mathcal{E}(A_1 : A_2) = \log \sum_{i,j,k,\ell} p_{ijk}^{1/2} p_{ij\ell}^{1/2} . \tag{29}$$

The sums over $k$ and $\ell$ can be performed separately, and we recast this result in the form

$$\mathcal{E}(A_1 : A_2) = \log \sum_{i,j} h_{ij}^2 , \tag{30a}$$

with

$$h_{ij} = \sum_k \left( \frac{\mathcal{Z}_{A_1}^{ik} \mathcal{Z}_{A_2}^{jk} \mathcal{Z}_B^{ij}}{\mathcal{Z}} \right)^{1/2} . \tag{30b}$$

Our calculations can straightforwardly be adapted to different geometries such as two imbricate squares.

As an important consistency check, the logarithmic negativity (30) must reduce to the known result for the Rényi entropy of index $n = 1/2$ [72] in the case where $A_1$ and $A_2$ are complementary subsystems. For $B = \emptyset$, the RK state (24) takes the form

$$|\psi\rangle = \sum_k \left( \frac{\mathcal{Z}_{A_1}^k \mathcal{Z}_{A_2}^k}{\mathcal{Z}} \right)^{1/2} |\psi_{A_1}^k\rangle \otimes |\psi_{A_2}^k\rangle . \tag{31}$$

Computing the logarithmic negativity in a similar manner as in the previous paragraphs, we find

$$\mathcal{E}(A_1 : A_2) = 2 \log \sum_k \left( \frac{\mathcal{Z}_{A_1}^k \mathcal{Z}_{A_2}^k}{\mathcal{Z}} \right)^{1/2} \tag{32}$$

$$= S_{1/2}(A_1) ,$$

in agreement with [72].

For local RK states, we expect the logarithmic negativity for adjacent regions to satisfy an area law, proportional to the area of the boundary shared by the two subsystems, as is observed for, e.g., two-dimensional topological systems [80, 81, 86] and free boson models [93, 94]. Indeed, for bipartite states with $B$ empty, the logarithmic negativity equals the 1/2–Rényi entropy, so if the Rényi entropies satisfy an area law—as, e.g., for dimer RK states on square and hexagonal lattices [72]—then the logarithmic negativity does too. Since the area law term is insensitive to the geometry, we further expect it to hold for logarithmic negativity of more general tripartitions with $B$ nonempty, the corresponding coefficient being also that of the 1/2–Rényi entropy.

For dimer RK states on the square lattice, one can be more quantitative. Let us consider the case where the three regions $A_1$, $A_2$ and $B$ are rectangles of sizes $L_X \times L$, with $X = A_1, A_2, B$, and $A_1, A_2$ share a common boundary of length $L$. The partition function $\mathcal{Z}_X$ of the dimer model on a $L_X \times L$ rectangle scales as [95]

$$\mathcal{Z}_X \sim e^{aL_X L - b_X L_X - bL + \cdots}, \tag{33}$$

where $a, b, b_X > 0$, and the ellipsis indicate subleading terms in the large-$L$, $L_X$ limit. Assuming that the fixed dimer configurations on the boundaries only affect the subleading coefficients $b, b_X$, but not the bulk coefficient $a$, we expect the probabilities $p_{ijk}$ in (25) to scale as $\log p_{ijk} \sim \alpha L + \dots$, with $\alpha > 0$ (see [72] for exact calculations for Rényi entropies). This in turn implies that the logarithmic negativity in (30) satisfies an area law, $\mathcal{E}(A_1 : A_2) \propto L$.

## 3 Resonating valence-bond states

In the context of lattice spin models, a valence bond is a spin singlet, and an RVB state is a quantum superposition of such valence bonds coverings, usually involving nearby spins. Schematically, a singlet can be represented as a dimer connecting two spins. Similarly to dimer RK states, RVB states with positive weights are thus constructed from an underlying classical dimer model, but the degrees of freedom are now spin-$S$ located on the vertices of the graph. The corresponding states are denoted SU($\mathcal{N}$) RVB state, with $\mathcal{N} = 2S + 1$ [51, 96, 97]. In the limit $\mathcal{N} \to \infty$, the valence-bond states become exactly orthogonal dimer states [51]. The results of this section are thus generalizations of those obtained in the previous one for RK states. In the following, we begin with SU(2) RVB states and study their separability and logarithmic negativity as a function of the distance $d$ between the subsystems. We discuss the case SU($\mathcal{N}$) in Sec. 3.7.

### 3.1 Definition for SU(2)

We work with the simplest RVB states, namely equal-weight superposition of spin-1/2 singlets, on arbitrary graphs. In our framework, singlets can be located on any edge of the graph. As such, nearest-neighbor and next to nearest-neighbor RVB states, for example, correspond to different underlying graphs. Since our results hold for arbitrary graphs, they encompass a wide variety of RVB states.

Given a spin-1/2 singlet configuration $\gamma$ of a given graph, the corresponding state $|\gamma\rangle$ is the product of singlets states between sites that are connected by a singlet,

$$|\gamma\rangle = \bigotimes_{(x,y)\in\gamma} |S_{x,y}\rangle, \tag{34}$$

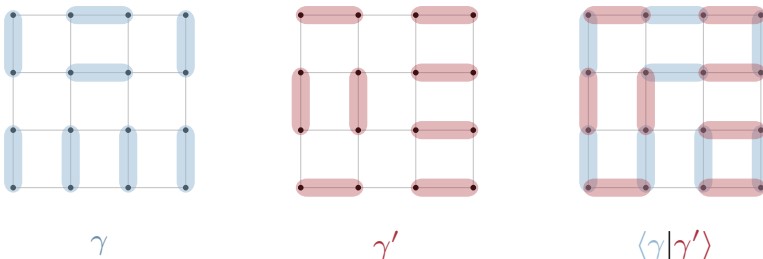

Figure 4: Two configurations, $\gamma$ and $\gamma'$, and the corresponding transition graph on the $4 \times 4$ square lattice. In this example, the number of sites is 16, the number of closed loops is 2, and therefore $\langle \gamma | \gamma' \rangle = 2^{-6}$.

where the notation $(x, y) \in \gamma$ indicates that the sites $x$ and $y$ are connected by a singlet in the configuration $\gamma$, and $|S_{x,y}\rangle$ is the spin-1/2 singlet state

$$|S_{x,y}\rangle = \frac{1}{\sqrt{2}}\left(|\uparrow_x \downarrow_y\rangle - |\downarrow_x \uparrow_y\rangle\right). \tag{35}$$

The states corresponding to different configurations $\gamma$ and $\gamma'$ are not orthogonal, and the value of the overlap $\langle \gamma | \gamma' \rangle$ can be read from the underlying singlet configurations. On the graph, one draws both configurations. The resulting image, denoted *transition graph*, consists of closed loops of singlets. We illustrate this in Fig. 4. The smallest loops have length two, when two singlets overlap. Denoting the number of closed loops by $n_\ell(\gamma, \gamma')$ and the number of sites on the graph by $N$, we have [98]

$$\langle \gamma | \gamma' \rangle = 2^{n_\ell(\gamma,\gamma') - N/2}. \tag{36}$$

For $\gamma = \gamma'$, this overlap is one since all the singlets perfectly overlap and the number of loops is exactly $N/2$.

The RVB state reads

$$|\Psi\rangle = \frac{1}{\sqrt{\mathcal{Z}}} \sum_{\gamma \in \Omega} |\gamma\rangle = \frac{1}{\sqrt{\mathcal{Z}}} \sum_{\gamma \in \Omega} \bigotimes_{(x,y) \in \gamma} |S_{x,y}\rangle, \tag{37}$$

where $\Omega$ denotes the set of all allowed singlet configurations on the graph, and $\mathcal{Z}$ is a constant that ensures $\langle \Psi | \Psi \rangle = 1$. From the overlap (36), it reads

$$\mathcal{Z} = \sum_{\gamma, \gamma' \in \Omega} 2^{n_\ell(\gamma,\gamma') - N/2}. \tag{38}$$

## 3.2 Tripartition and disconnected subsystems

Let us consider a tripartition $A_1 \cup B \cup A_2$ of the graph. Each subsystem consists in a set of $N_{A_1}$, $N_B$ and $N_{A_2}$ vertices, respectively. By definition, a boundary site belonging to a subsystem is connected through an edge to at least one site from a different subsystem. Similarly, boundary edges are edges of the graph that connect sites from different subsystems. Importantly, we assume that $A_1$ and $A_2$ are disconnected, namely there are no boundary edges that connect sites in $A_1$ to $A_2$. The distance $d$ between $A_1$ and $A_2$ is defined as the minimal number of edges needed to connect two boundary sites in $B$, pertaining to different boundaries. We illustrate such a tripartition in Fig. 5 for the square lattice.

Our goal is to express the RVB state (37) in terms of RVB states for each subsystem. We denote by $\Omega_{\text{bd}}^k$, $k = 1, 2$, the set of allowed singlet configurations on boundary edges that

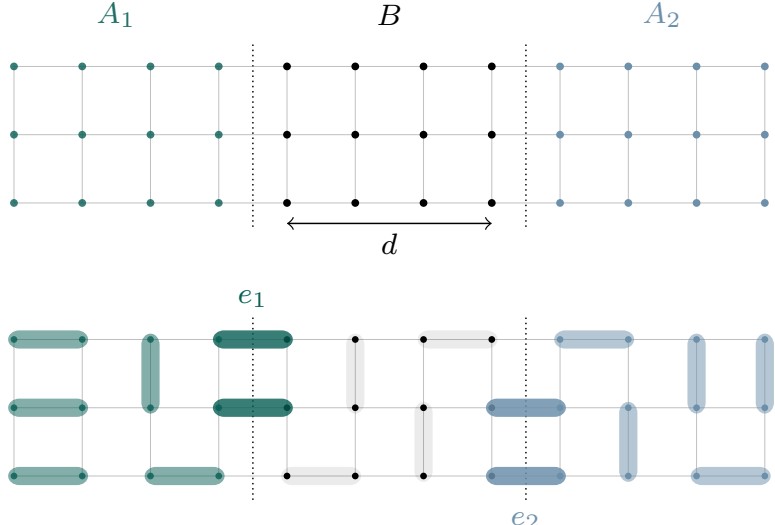

Figure 5: *Top*: Example of a tripartition for the RVB state on a $3 \times 12$ square lattice. Here, $N_{A_1} = N_{A_2} = N_B = 12$ and $d = 3$. *Bottom*: A singlet configuration on the same lattice as in the top panel. The boundary singlets in $e_1$ and $e_2$ are highlighted.

connect sites in $A_k$ to $B$. Singlet states defined on boundary edges are called boundary singlets. Given two boundary configurations $e_1, e_2$ in $\Omega_{\mathrm{bd}}^1$ and $\Omega_{\mathrm{bd}}^2$, respectively, we define $\Omega^{e_1, e_2}$ as the set of all singlet configurations on the whole graph, from which we removed all the edges connected to occupied boundary sites in $e_1, e_2$. We give an example of a singlet configuration in Fig. 5.

We can recast the RVB state (37) as

$$|\Psi\rangle = \frac{1}{\sqrt{\mathcal{Z}}} \sum_{e_1 \in \Omega_{\mathrm{bd}}^1} \sum_{e_2 \in \Omega_{\mathrm{bd}}^2} \sum_{\gamma \in \Omega^{e_1, e_2}} |e_1\rangle \otimes |\gamma\rangle \otimes |e_2\rangle, \tag{39}$$

where $|e_k\rangle$, $k = 1, 2$, is the product of boundary singlet in the boundary configuration $e_k$,

$$|e_k\rangle = \bigotimes_{(i_k, j_k) \in e_k} |S_{i_k, j_k}\rangle. \tag{40}$$

By convention, the sites $i_k$ belong to $A_k$, whereas $j_k$ label sites in $B$. By abuse of notation, we will sometimes write $i_k \in e_k$ to denote the sites in $A_k$ that are occupied by a boundary singlet in $e_k$, and $j_k \in e_k$ to denote the corresponding sites in $B$. We further introduce $\Omega_{A_k}^{e_k}$, $k = 1, 2$, as the set of singlet configurations on the system $A_k$ from which we removed the edges connected to an occupied site in the boundary configuration $e_k$. We also introduce $\Omega_B^{e_1, e_2}$, which is the equivalent quantity for system $B$, and it depends on both boundary configurations $e_1, e_2$. With these notations, we have

$$\sum_{\gamma \in \Omega^{e_1, e_2}} |\gamma\rangle = \sum_{\gamma_{A_1} \in \Omega_{A_1}^{e_1}} \sum_{\gamma_B \in \Omega_B^{e_1, e_2}} \sum_{\gamma_{A_2} \in \Omega_{A_2}^{e_2}} |\gamma_{A_1}\rangle \otimes |\gamma_B\rangle \otimes |\gamma_{A_2}\rangle, \tag{41}$$

where $|\gamma_X\rangle$, $X = A_1, B, A_2$, are defined as in (34). Finally, we introduce

$$|\Psi_{A_k}^{e_k}\rangle \equiv \frac{1}{\sqrt{\mathcal{Z}_{A_k}^{e_k}}} \sum_{\gamma_{A_k} \in \Omega_{A_k}^{e_k}} |\gamma_{A_k}\rangle,$$

$$|\Psi_B^{e_1, e_2}\rangle \equiv \frac{1}{\sqrt{\mathcal{Z}_B^{e_1, e_2}}} \sum_{\gamma_B \in \Omega_B^{e_1, e_2}} |\gamma_B\rangle, \tag{42}$$

where

$$\mathcal{Z}_{A_k}^{e_k} = \sum_{\gamma_{A_k}, \gamma'_{A_k} \in \Omega_{A_k}^{e_k}} 2^{n_\ell(\gamma_{A_k}, \gamma'_{A_k}) - N_{A_k}/2},$$

$$\mathcal{Z}_B^{e_1, e_2} = \sum_{\gamma_B, \gamma'_B \in \Omega_B^{e_1, e_2}} 2^{n_\ell(\gamma_B, \gamma'_B) - N_B/2}, \tag{43}$$

and rewrite (39) as

$$|\Psi\rangle = \sum_{e_1 \in \Omega_{\text{bd}}^1} \sum_{e_2 \in \Omega_{\text{bd}}^2} \left( \frac{\mathcal{Z}_{A_1}^{e_1} \mathcal{Z}_{A_2}^{e_2} \mathcal{Z}_B^{e_1, e_2}}{\mathcal{Z}} \right)^{1/2} |\Psi_{A_1}^{e_1}\rangle \otimes |e_1\rangle \otimes |\Psi_B^{e_1, e_2}\rangle \otimes |e_2\rangle \otimes |\Psi_{A_2}^{e_2}\rangle. \tag{44}$$

## 3.3 Reduced density matrix

As the degrees of freedom reside on the vertices of the graph, we compute the reduced density matrix as

$$\rho_{A_1 \cup A_2} = \sum_{\substack{\sigma_j = \uparrow, \downarrow \\ j \in B}} \langle \sigma_1 \cdots \sigma_{N_B} | \Psi \rangle \langle \Psi | \sigma_1 \cdots \sigma_{N_B} \rangle, \tag{45}$$

where the sum is over all the spin configurations in $B$. From (44), we find

$$\rho_{A_1 \cup A_2} = \sum_{e_1, e'_1 \in \Omega_{\text{bd}}^1} \sum_{e_2, e'_2 \in \Omega_{\text{bd}}^2} \left( \frac{\mathcal{Z}_{A_1}^{e_1} \mathcal{Z}_{A_2}^{e_2} \mathcal{Z}_B^{e_1, e_2}}{\mathcal{Z}} \right)^{1/2} \left( \frac{\mathcal{Z}_{A_1}^{e'_1} \mathcal{Z}_{A_2}^{e'_2} \mathcal{Z}_B^{e'_1, e'_2}}{\mathcal{Z}} \right)^{1/2} |\Psi_{A_1}^{e_1}\rangle \langle \Psi_{A_1}^{e'_1}| \otimes |\Psi_{A_2}^{e_2}\rangle \langle \Psi_{A_2}^{e'_2}|$$

$$\times \sum_{\substack{\sigma_j = \uparrow, \downarrow \\ j \in B}} \langle \sigma_1 \cdots \sigma_{N_B} | \left( |e_1\rangle \otimes |\Psi_B^{e_1, e_2}\rangle \otimes |e_2\rangle \right) \left( \langle e'_2| \otimes \langle \Psi_B^{e'_1, e'_2}| \otimes \langle e'_1| \right) |\sigma_1 \cdots \sigma_{N_B}\rangle. \tag{46}$$

We recall that the state $|e_1\rangle$, for instance, is a product of singlets that involve boundary sites in $B$ and in $A_1$. In the sum over the spin values $\sigma_j = \uparrow, \downarrow$ for boundary sites in $B$ occupied by a boundary singlet in the configurations $\{e_1, e_2, e'_1, e'_2\}$, the corresponding spins in $A_1$ or $A_2$ are thus fixed to be of opposite value.

For $\sigma = \uparrow, \downarrow$, we define $\bar{\sigma} = \downarrow, \uparrow$, and we introduce the notations

$$|\boldsymbol{\sigma}_{e_1}\rangle = \bigotimes_{j \in e_1} |\sigma_j\rangle_B,$$

$$|\bar{\boldsymbol{\sigma}}_{e_1}\rangle = \bigotimes_{j \in e_1} |\bar{\sigma}_j\rangle_{A_1}, \tag{47}$$

for a given spin configuration $\{\sigma_j\}$, $j \in e_1$ of occupied boundary sites in $B$, and similarly for $e_2$. We stress that the product in the first line of (47) is over the sites in $B$ that are occupied by a boundary singlet in the configuration $e_1$, whereas the product on the second line is over the corresponding sites in $A_1$, as highlighted by the notation in the right-hand side of (47). After some algebra, we arrive at

$$\rho_{A_1 \cup A_2} = \sum_{e_1, e'_1 \in \Omega_{\text{bd}}^1} \sum_{e_2, e'_2 \in \Omega_{\text{bd}}^2} \sum_{\substack{\sigma_j = \uparrow, \downarrow \\ j \in \{e_1, e'_1, e_2, e'_2\}}} 2^{-\frac{1}{2}|\{e_1, e_2, e'_1, e'_2\}|} \left( \frac{\mathcal{Z}_{A_1}^{e_1} \mathcal{Z}_{A_2}^{e_2} \mathcal{Z}_B^{e_1, e_2}}{\mathcal{Z}} \right)^{1/2} \left( \frac{\mathcal{Z}_{A_1}^{e'_1} \mathcal{Z}_{A_2}^{e'_2} \mathcal{Z}_B^{e'_1, e'_2}}{\mathcal{Z}} \right)^{1/2}$$

$$\times \langle \Psi_B^{e'_1, e'_2} \otimes \boldsymbol{\sigma}_{e'_1} \otimes \boldsymbol{\sigma}_{e'_2} | \Psi_B^{e_1, e_2} \otimes \boldsymbol{\sigma}_{e_1} \otimes \boldsymbol{\sigma}_{e_2} \rangle \left( |\Psi_{A_1}^{e_1} \otimes \bar{\boldsymbol{\sigma}}_{e_1}\rangle \langle \Psi_{A_1}^{e'_1} \otimes \bar{\boldsymbol{\sigma}}_{e'_1}| \right) \otimes \left( |\Psi_{A_2}^{e_2} \otimes \bar{\boldsymbol{\sigma}}_{e_2}\rangle \langle \Psi_{A_2}^{e'_2} \otimes \bar{\boldsymbol{\sigma}}_{e'_2}| \right), \tag{48}$$

where $|\{e_1, e_2, e'_1, e'_2\}|$ is the number of boundary singlets in the combined configurations $\{e_1, e_2, e'_1, e'_2\}$, and factor of $1/2$ originates from the singlet normalization.

For simplicity, we write the reduced density matrix as

$$
\rho_{A_1 \cup A_2} = \sum_{e_1, e'_1 \in \Omega^1_{\text{bd}}} \sum_{\substack{e_2, e'_2 \in \Omega^2_{\text{bd}}}} \sum_{\substack{\sigma_j = \uparrow, \downarrow \\ j \in \{e_1, e'_1, e_2, e'_2\}}} \mathcal{F}(e_1, e_2; e'_1, e'_2; \boldsymbol{\sigma}_{\text{bd}})
$$
$$
\times \left( |\Psi^{e_1}_{A_1} \otimes \bar{\boldsymbol{\sigma}}_{e_1} \rangle \langle \Psi^{e'_1}_{A_1} \otimes \bar{\boldsymbol{\sigma}}_{e'_1} | \right) \otimes \left( |\Psi^{e_2}_{A_2} \otimes \bar{\boldsymbol{\sigma}}_{e_2} \rangle \langle \Psi^{e'_2}_{A_2} \otimes \bar{\boldsymbol{\sigma}}_{e'_2} | \right), \quad (49)
$$

with

$$
\mathcal{F}(e_1, e_2; e'_1, e'_2; \boldsymbol{\sigma}_{\text{bd}}) = 2^{-\frac{1}{2}|\{e_1, e_2, e'_1, e'_2\}|} \left( \frac{\mathcal{Z}^{e_1}_{A_1} \mathcal{Z}^{e_2}_{A_2}}{\mathcal{Z}} \right)^{1/2} \left( \frac{\mathcal{Z}^{e'_1}_{A_1} \mathcal{Z}^{e'_2}_{A_2}}{\mathcal{Z}} \right)^{1/2}
$$
$$
\times \left( \mathcal{Z}^{e_1, e_2}_B \mathcal{Z}^{e'_1, e'_2}_B \right)^{1/2} \langle \Psi^{e'_1, e'_2}_B \otimes \boldsymbol{\sigma}_{e'_1} \otimes \boldsymbol{\sigma}_{e'_2} | \Psi^{e_1, e_2}_B \otimes \boldsymbol{\sigma}_{e_1} \otimes \boldsymbol{\sigma}_{e_2} \rangle, \quad (50)
$$

where $\boldsymbol{\sigma}_{\text{bd}} \equiv \{\sigma_j\}$, $j \in \{e_1, e_2, e'_1, e'_2\}$ is the spin configuration of occupied boundary sites in $B$.

## 3.4 Overlap

Let us introduce the notation

$$
\mathcal{G}(e_1, e_2; e'_1, e'_2; \boldsymbol{\sigma}_{\text{bd}}) \equiv \left( \mathcal{Z}^{e_1, e_2}_B \mathcal{Z}^{e'_1, e'_2}_B \right)^{1/2} \langle \Psi^{e'_1, e'_2}_B \otimes \boldsymbol{\sigma}_{e'_1} \otimes \boldsymbol{\sigma}_{e'_2} | \Psi^{e_1, e_2}_B \otimes \boldsymbol{\sigma}_{e_1} \otimes \boldsymbol{\sigma}_{e_2} \rangle, \quad (51)
$$

and describe how to compute this normalized overlap graphically, in a similar fashion as for the overlap in (36). In the following, we use the lighter notation $\mathcal{G}(e_1, e_2; e'_1, e'_2) \equiv \mathcal{G}(e_1, e_2; e'_1, e'_2; \boldsymbol{\sigma}_{\text{bd}})$, but one should keep in mind that $\mathcal{G}(e_1, e_2; e'_1, e'_2)$ does not only depend on the boundary singlet configurations, but also on the corresponding boundary spin configuration $\boldsymbol{\sigma}_{\text{bd}}$. We will use the same notation for $\mathcal{F}(e_1, e_2; e'_1, e'_2)$.

The overlap in $\mathcal{G}(e_1, e_2; e'_1, e'_2)$ involves a double sum over configurations $\gamma_B$ and $\gamma'_B$, see (42). First, we isolate one term in the double sum, and focus on the overlap $\langle \gamma'_B \otimes \boldsymbol{\sigma}_{e'_1} \otimes \boldsymbol{\sigma}_{e'_2} | \gamma_B \otimes \boldsymbol{\sigma}_{e_1} \otimes \boldsymbol{\sigma}_{e_2} \rangle$. One draws the fixed boundary spins $\boldsymbol{\sigma}_{\text{bd}}$, and the singlets of the configurations $\gamma_B$ and $\gamma'_B$ on the same graph. In the resulting transition graph, fixed spins are connected by strings of singlets, and in the rest of the domain there are closed singlet loops. It is convenient to draw the spins and singlets of the bra $\langle \gamma'_B \otimes \boldsymbol{\sigma}_{e'_1} \otimes \boldsymbol{\sigma}_{e'_2} |$ in red, and those of the ket $|\gamma_B \otimes \boldsymbol{\sigma}_{e_1} \otimes \boldsymbol{\sigma}_{e_2} \rangle$ in blue.

Because singlets have zero magnetisation, we have the following rules: (i) two fixed boundary spins of the same color can be connected by a string of singlets only if they are opposite, whereas (ii) two fixed boundary spins with different colors can only be connected if they are equal. If those rules are not satisfied, the bra and ket involved have different magnetisation, and hence the resulting overlap is zero. We illustrate this graphical construction in Fig. 6.

To compute the overlap from the transition graph, we generalize (36) to account for the presence of singlet strings. A string of $n_D$ singlets connecting two fixed spins has weight $2^{-n_D/2}$, irrespective of the colors or orientation of the fixed boundary spins, provided that the connectivity rules from the previous paragraph are satisfied.

Let $\Gamma = \{\gamma_B, e_1, e_2, \boldsymbol{\sigma}_{\text{bd}}\}$ encode all the information about the configuration $\gamma_B$, the boundary singlets and boundary spins configurations. To proceed, we need to introduce four additional notations: (a) the total number of strings is $n_s(\Gamma, \Gamma')$, (b) the total number of singlets in

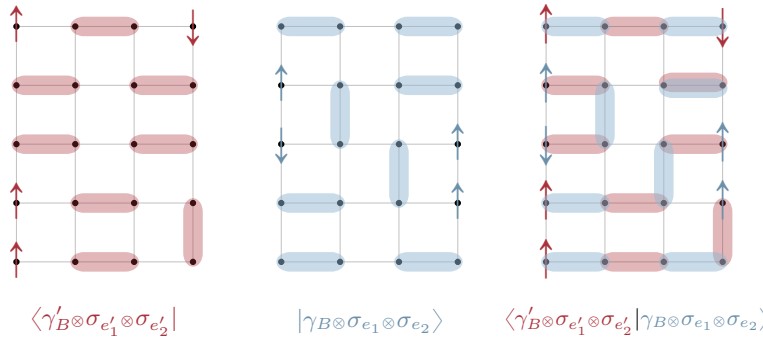

$$\langle \gamma'_B \otimes \sigma_{e'_1} \otimes \sigma_{e'_2}| \qquad |\gamma_B \otimes \sigma_{e_1} \otimes \sigma_{e_2}\rangle \qquad \langle \gamma'_B \otimes \sigma_{e'_1} \otimes \sigma_{e'_2}|\gamma_B \otimes \sigma_{e_1} \otimes \sigma_{e_2}\rangle$$

Figure 6: Illustration of the graphic method to compute the overlap $\langle \gamma'_B \otimes \boldsymbol{\sigma}_{e'_1} \otimes \boldsymbol{\sigma}_{e'_2}|\gamma_B \otimes \boldsymbol{\sigma}_{e_1} \otimes \boldsymbol{\sigma}_{e_2}\rangle$ on a $4 \times 5$ domain. The fixed boundary spins are illustrated by arrows.

the strings is $n_D(\Gamma, \Gamma')$, and (c) the number of closed singlet loops is $n_\ell(\Gamma, \Gamma')$. Moreover, (d) the number of sites that are not in a string of singlets is

$$\tilde{N}_B(\Gamma, \Gamma') = N_B - \left(n_D(\Gamma, \Gamma') + n_s(\Gamma, \Gamma')\right). \tag{52}$$

With these conventions, the overlap is

$$\langle \gamma'_B \otimes \boldsymbol{\sigma}_{e'_1} \otimes \boldsymbol{\sigma}_{e'_2}|\gamma_B \otimes \boldsymbol{\sigma}_{e_1} \otimes \boldsymbol{\sigma}_{e_2}\rangle = 2^{-n_D(\Gamma, \Gamma')/2} 2^{n_\ell(\Gamma, \Gamma') - \tilde{N}_B(\Gamma, \Gamma')/2}, \tag{53}$$

where the first factor arises from the singlet strings contributions, and the second comes from the closed singlet loops contributions, as in (36). Simplifying this expression, the result for the total overlap $\mathcal{G}(e_1, e_2; e'_1, e'_2)$ is

$$\mathcal{G}(e_1, e_2; e'_1, e'_2) = \sum_{\gamma_B \in \Omega_B^{e_1, e_2}} \sum_{\gamma'_B \in \Omega_B^{e'_1, e'_2}} 2^{n_\ell(\Gamma, \Gamma') - (N_B - n_s(\Gamma, \Gamma'))/2}. \tag{54}$$

### 3.5 Separability for disconnected subsystems

In this section, we show that the reduced density matrix (49) is separable, up to exponentially small terms in the distance $d$ between $A_1$ and $A_2$. Our argument is twofold. First, we show that the reduced density matrix satisfies $\rho_{A_1 \cup A_2}^{T_1} = \rho_{A_1 \cup A_2}$ up to exponentially small terms in $d$. Second, we argue that the symmetric part of the reduced density matrix can be written in the separable form of (1).

### 3.5.1 Symmetry under partial transpose

In what follows, we show

$$\mathcal{F}(e_1, e_2; e'_1, e'_2) = \mathcal{F}(e'_1, e_2; e_1, e'_2) + \mathcal{O}(2^{-d/2}), \tag{55}$$

implying that $\rho_{A_1 \cup A_2}$ in (49) is symmetric under partial transposition, up to exponentially small terms in $d$.

Crucially, we note that $\mathcal{G}(e_1, e_2; e'_1, e'_2)$ (and thus $\mathcal{F}(e_1, e_2; e'_1, e'_2)$) vanishes, unless

$$m(e_1) + m(e_2) = m(e'_1) + m(e'_2), \tag{56}$$

where $m(e) \equiv \sum_{j \in e} \sigma_j$ is the total magnetisation of the fixed boundary spins in $B$ occupied by boundary singlets in the configuration $e$. This holds because $|\Psi_B^{e_1, e_2}\rangle$ and $|\Psi_B^{e'_1, e'_2}\rangle$ are states with zero magnetisation and the overlap (51) is zero, unless the magnetisation in the bra and the ket are equal. This is exactly condition (56).

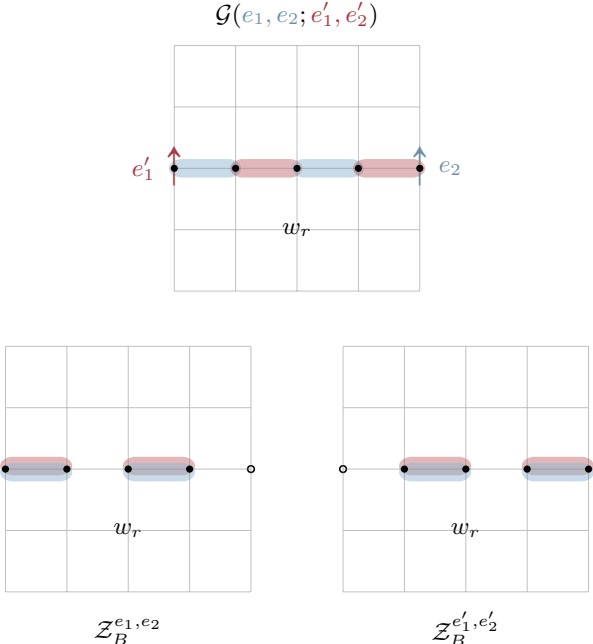

Figure 7: Each transition graph in $\mathcal{G}(e_1, e_2; e_1', e_2')$ has at least one string of length $d$ or larger, and the rest of the configuration has weight $w_r$. For each such transition graph, there is a transition graph in $\mathcal{Z}_B^{e_1, e_2}$ and $\mathcal{Z}_B^{e_1', e_2'}$ where the string is replaced by overlapping singlets with weight one, and the whole configuration has weight $w_r$.

**The case $m(e_1) \neq m(e_1')$.** Boundary configurations such that $\mathcal{G}(e_1, e_2; e_1', e_2') \neq 0$ but $\mathcal{G}(e_1', e_2; e_1, e_2') = 0$, can break the invariance under the exchange $e_1 \leftrightarrow e_1'$. This happens if (56) holds, but

$$m(e_1') + m(e_2) \neq m(e_1) + m(e_2'), \tag{57}$$

namely if $m(e_1) \neq m(e_1')$. In that case, with the rules for the connectivity of fixed spins described in Sec. 3.4, one can show that each transition graph that appears in the normalized overlap $\mathcal{G}(e_1, e_2; e_1', e_2')$ contains at least one string of singlets that stretches across $B$ and connects boundary sites adjacent to $A_1$ and $A_2$.

We recall that, by definition, the minimal distance between two boundary points in $B$ pertaining to different boundaries is $d$, and hence $n_D(\Gamma, \Gamma') \geqslant d$. Moreover, the total number of strings satisfies $n_s(\Gamma, \Gamma') = |\{e_1, e_2, e_1', e_2'\}|/2$ and is thus fixed by the boundary-singlet configurations, but does not depend on the magnetisation. Hence, the number of closed singlet loops in each transition graph is bounded form above,

$$n_\ell(\Gamma, \Gamma') \leqslant \frac{N_B - (d + n_s(\Gamma, \Gamma'))}{2}. \tag{58}$$

The bound is saturated if there is only one string of singlets, of minimal length $d$, and that all other singlets perfectly overlap, hence maximizing the number of loops. As a consequence of (58), each term in the sum in (54) is of order $2^{-d/2}$ or smaller. However, this bound is not enough to conclude that (55) holds for $m(e_1) \neq m(e_1')$, because there is an exponential number of terms in the sum in (54) which could add up to cancel the individual exponential suppression of each term. We thus develop our arguments to show that $\mathcal{F}(e_1, e_2; e_1', e_2')$ in (50) is negligible for $m(e_1) \neq m(e_1')$.

First, we note that

$$\left( \frac{\mathcal{Z}_{A_1}^{e_1} \mathcal{Z}_{A_2}^{e_2} \mathcal{Z}_B^{e_1,e_2}}{\mathcal{Z}} \right) \leqslant 1 \,, \tag{59}$$

and hence

$$\mathcal{F}(e_1,e_2;e_1',e_2') \leqslant \frac{\mathcal{G}(e_1,e_2;e_1',e_2')}{(\mathcal{Z}_B^{e_1,e_2} \mathcal{Z}_B^{e_1',e_2'})^{1/2}} \,. \tag{60}$$

The numerator $\mathcal{G}(e_1,e_2;e_1',e_2')$ is a sum over $\gamma_B \in \Omega_B^{e_1,e_2}$ and $\gamma_B' \in \Omega_B^{e_1',e_2'}$. For each choice of $\gamma_B, \gamma_B'$, the transition graph has at least one string of length $d$ or larger. The total weight of the strings thus satisfies $w_s(\gamma_B,\gamma_B') = \mathcal{O}(2^{-d/2})$, and the weight of the rest of the transition graph from which the strings are excluded is $w_r(\gamma_B,\gamma_B') \leqslant 1$. We thus have

$$\begin{aligned}
\mathcal{G}(e_1,e_2;e_1',e_2') &= \sum_{\gamma_B \in \Omega_B^{e_1,e_2}} \sum_{\gamma_B' \in \Omega_B^{e_1',e_2'}} w_s(\gamma_B,\gamma_B') w_r(\gamma_B,\gamma_B') \\
&= \mathcal{O}(2^{-d/2}) \left( \sum_{\gamma_B \in \Omega_B^{e_1,e_2}} \sum_{\gamma_B' \in \Omega_B^{e_1',e_2'}} w_r(\gamma_B,\gamma_B') \right) .
\end{aligned} \tag{61}$$

Second, we turn to the investigation of the denominator in the right-hand side of (60). Similarly to $\mathcal{G}(e_1,e_2;e_1',e_2')$, the partition functions $\mathcal{Z}_B^{e_1,e_2}$ and $\mathcal{Z}_B^{e_1',e_2'}$ are also sums over transition graphs, see (43). For each transition graph in $\mathcal{G}(e_1,e_2;e_1',e_2')$, there is one transition graph in $\mathcal{Z}_B^{e_1,e_2}$ where the strings are replaced by overlapping singlets with weight one and the rest of the configuration is identical, with weight $w_r(\gamma_B,\gamma_B')$. The same argument holds for $\mathcal{Z}_B^{e_1',e_2'}$. We illustrate this in Fig. 7. Moreover, both partition functions contain more terms than those described here. Hence, we have

$$(\mathcal{Z}_B^{e_1,e_2} \mathcal{Z}_B^{e_1',e_2'})^{1/2} \geqslant \sum_{\gamma_B \in \Omega_B^{e_1,e_2}} \sum_{\gamma_B' \in \Omega_B^{e_1',e_2'}} w_r(\gamma_B,\gamma_B') \,. \tag{62}$$

Finally, combining equations (60), (61) and (62) we conclude that $\mathcal{F}(e_1,e_2;e_1',e_2') = \mathcal{O}(2^{-d/2})$ and hence (55) holds for $m(e_1) \neq m(e_1')$.

**The case $m(e_1) = m(e_1')$.** To show separability up to exponentially small corrections, it remains to show that (55) holds when

$$m(e_1) + m(e_2) = m(e_1') + m(e_2') \,, \tag{63}$$

and

$$m(e_1') + m(e_2) = m(e_1) + m(e_2') \,, \tag{64}$$

that is if $m(e_1) = m(e_1')$. In that case, both $\mathcal{G}(e_1,e_2;e_1',e_2')$ and $\mathcal{G}(e_1',e_2;e_1,e_2')$ are non-vanishing. Again, our arguments use the fact that $\mathcal{G}(e_1,e_2;e_1',e_2')$ is a sum over transition graphs. In the sum, there are two distinct types of transition graphs: (I) those without strings that connect different boundaries, and (II) those with at least one string that stretches across $B$ to connect different boundaries.

For graphs of type I, there are nonetheless singlet strings, but they only connect boundary sites pertaining to the same boundary. For each such graph in $\mathcal{G}(e_1,e_2;e_1',e_2')$, there is a graph with the exact same weight in $\mathcal{G}(e_1',e_2;e_1,e_2')$ where each string attached to the boundary between $A_1$ and $B$ is drawn in opposite colors. We illustrate this in the top panel of Fig. 8. If

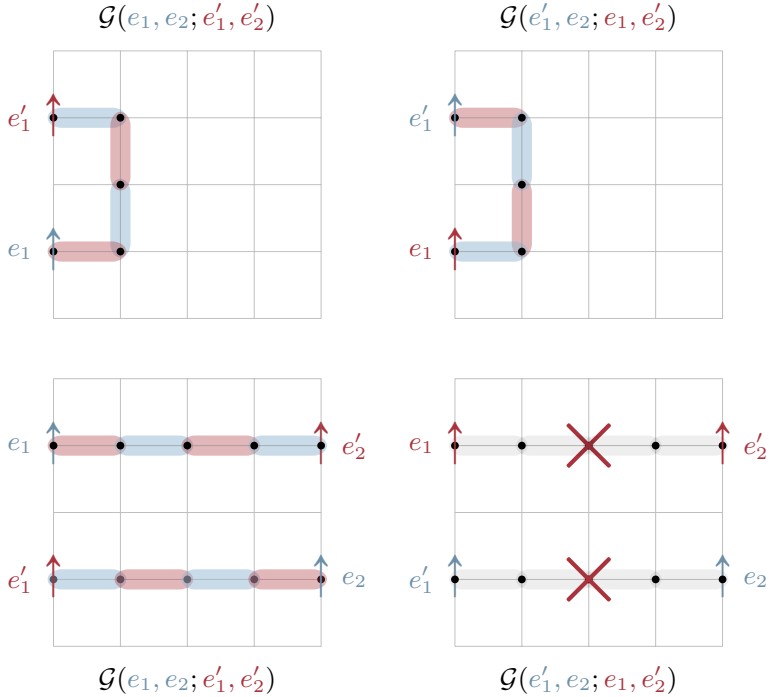

Figure 8: *Top panels:* For each transition graphs in $\mathcal{G}(e_1, e_2; e_1', e_2')$ where no singlet strings connect both boundaries, there is a transition graph in $\mathcal{G}(e_1', e_2; e_1, e_2')$ with the same weight, where the singlet strings pertaining to the boundary between $A_1$ and $B$ have opposite colors. *Bottom panels:* For each transition graphs in $\mathcal{G}(e_1, e_2; e_1', e_2')$ where at least one singlet string connects both boundaries, there is no counterpart in $\mathcal{G}(e_1', e_2; e_1, e_2')$ because of coloring arguments. However, these configurations are exponentially suppressed, as discussed in the previous paragraphs.

it were not for type-II graphs, we would thus have a perfect equality between $\mathcal{G}(e_1, e_2; e_1', e_2')$ and $\mathcal{G}(e_1', e_2; e_1, e_2')$.

For graphs of type II, the above pictorial argument does not work. Since we consider partial transposition with respect to $A_1$, we draw the boundary spins along $A_1$ in a different color in $\mathcal{G}(e_1, e_2; e_1', e_2')$ and $\mathcal{G}(e_1', e_2; e_1, e_2')$, whereas those at the boundary with $A_2$ are identical in both overlaps. Hence, if a string connects boundary spins from different boundaries in $\mathcal{G}(e_1, e_2; e_1', e_2')$, the configuration where a spin along the boundary of $A_1$ is drawn in opposite color is forbidden and has weight zero. We illustrate this in the bottom panel of Fig. 8. Those transition graphs thus break the symmetry $e_1 \leftrightarrow e_1'$. However, each such transition graph has at least one string of length greater than $d$, with weight $w_s = \mathcal{O}(2^{-d/2})$. Using similar arguments as for the case $m(e_1) \neq m(e_1')$, we can argue that the correction due to type-II graphs is exponentially small in $d$. We thus conclude that (55) holds for $m(e_1) = m(e_1')$, and in general.

### 3.5.2 Separable form of the reduced density matrix

In the previous section, we have established that the reduced density matrix $\rho_{A_1 \cup A_2}$ takes the form

$$\rho_{A_1 \cup A_2} = \rho_{A_1 \cup A_2}^{\mathrm{s}} + \tilde{\rho}_{A_1 \cup A_2}, \tag{65}$$

where $\rho_{A_1 \cup A_2}^{\mathrm{s}}$ is the symmetric part of the matrix satisfying $(\rho_{A_1 \cup A_2}^{\mathrm{s}})^{T_1} = \rho_{A_1 \cup A_2}^{\mathrm{s}}$. The second term $\tilde{\rho}_{A_1 \cup A_2}$ breaks the invariance under partial transposition, but its matrix elements are of

order $2^{-d/2}$. Now, we prove the stronger statement that $\rho^{\mathrm{s}}_{A_1 \cup A_2}$ is separable as in (1).

We start with

$$
\rho^{\mathrm{s}}_{A_1 \cup A_2} = \sum_{e_1, e'_1 \in \Omega^1_{\mathrm{bd}}} \sum_{e_2, e'_2 \in \Omega^2_{\mathrm{bd}}} \sum_{\substack{\sigma_j = \uparrow, \downarrow \\ j \in \{e_1, e'_1, e_2, e'_2\}}} \mathcal{F}^{\mathrm{s}}(e_1, e_2; e'_1, e'_2)
$$
$$
\times \left( |\Psi^{e_1}_{A_1} \otimes \bar{\boldsymbol{\sigma}}_{e_1}\rangle \langle \Psi^{e'_1}_{A_1} \otimes \bar{\boldsymbol{\sigma}}_{e'_1}| \right) \otimes \left( |\Psi^{e_2}_{A_2} \otimes \bar{\boldsymbol{\sigma}}_{e_2}\rangle \langle \Psi^{e'_2}_{A_2} \otimes \bar{\boldsymbol{\sigma}}_{e'_2}| \right), \quad (66)
$$

where $\mathcal{F}^{\mathrm{s}}(e_1, e_2; e'_1, e'_2)$ only contains terms and transition graphs that are invariant under $e_1 \leftrightarrow e'_1$ (and $e_2 \leftrightarrow e'_2$). In particular, every term in the sum satisfies $m(e_1) = m(e'_1)$ and $m(e_2) = m(e'_2)$. We recast (66) as

$$
\rho^{\mathrm{s}}_{A_1 \cup A_2} = \sum_{e_1, e'_1 \in \Omega^1_{\mathrm{bd}}} \sum_{e_2, e'_2 \in \Omega^2_{\mathrm{bd}}} \sum_{\substack{\sigma_j = \uparrow, \downarrow \\ j \in \{e_1, e'_1, e_2, e'_2\}}} \mathcal{F}^{\mathrm{s}}(e_1, e_2; e'_1, e'_2) \mathcal{Z}^{e_1, e'_1}_{A_1} \mathcal{Z}^{e_2, e'_2}_{A_2} \left( \rho^{e_1, e'_1}_{A_1} \otimes \rho^{e_2, e'_2}_{A_2} \right), \quad (67a)
$$

with

$$
\rho^{e_k, e'_k}_{A_k} = \frac{1}{2 \mathcal{Z}^{e_k, e'_k}_{A_k}} \left( |\Psi^{e_k}_{A_k} \otimes \bar{\boldsymbol{\sigma}}_{e_k}\rangle \langle \Psi^{e'_k}_{A_k} \otimes \bar{\boldsymbol{\sigma}}_{e'_k}| + |\Psi^{e'_k}_{A_k} \otimes \bar{\boldsymbol{\sigma}}_{e'_k}\rangle \langle \Psi^{e_k}_{A_k} \otimes \bar{\boldsymbol{\sigma}}_{e_k}| \right), \quad (67b)
$$

and

$$
Z^{e_k, e'_k}_{A_k} = \langle \Psi^{e'_k}_{A_k} \otimes \bar{\boldsymbol{\sigma}}_{e'_k} | \Psi^{e_k}_{A_k} \otimes \bar{\boldsymbol{\sigma}}_{e_k} \rangle. \quad (67c)
$$

The normalization $Z^{e_k, e'_k}_{A_k}$, $k = 1, 2$, is non-zero since $m(e_k) = m(e'_k)$, such that the magnetization of both terms in the overlap is equal. The density matrices $\rho^{e_k, e'_k}_{A_k}$ are thus well-defined Hermitian operators with unit trace. The operator $\rho^{\mathrm{s}}_{A_1 \cup A_2}$ in (67a) is thus separable.

## 3.6 Logarithmic negativity

Here we investigate the logarithmic negativity of disjoint subsystems in SU(2) RVB states on arbitrary graphs. We consider the quantity

$$
\begin{aligned}
\mathcal{T}_{2n} &\equiv \frac{\mathrm{Tr}(\rho^{T_1}_{A_1 \cup A_2})^{2n} - \mathrm{Tr}(\rho_{A_1 \cup A_2})^{2n}}{\mathrm{Tr}(\rho_{A_1 \cup A_2})^{2n}} \\
&= \frac{\mathrm{Tr}\left[ (\rho^{T_1}_{A_1 \cup A_2})^{2n} - (\rho_{A_1 \cup A_2})^{2n} \right]}{\mathrm{Tr}(\rho_{A_1 \cup A_2})^{2n}},
\end{aligned} \quad (68)
$$

for integer $n$. Importantly, in the limit $n \to 1/2$, we have $\mathcal{T}_1 = \mathrm{Tr}|\rho^{T_1}_{A_1 \cup A_2}| - 1$. Using the result of the previous section, the numerator is a sum of terms each of order $2^{-d/2}$ at most. The denominator prevents the sum in the numerator to cancel the exponential suppression of the individual terms, and we find

$$
\mathcal{T}_{2n} = \mathcal{O}(2^{-d/2}), \quad (69)
$$

irrespective of $n$. Taking the limit $n \to 1/2$, we obtain

$$
\mathrm{Tr}|\rho^{T_1}_{A_1 \cup A_2}| = 1 + \mathcal{T}_1 = 1 + \mathcal{O}(2^{-d/2}), \quad (70)
$$

or, equivalently,

$$
\mathcal{E}(A_1 : A_2) = \mathcal{O}(2^{-d/2}), \quad (71)
$$

where we used the replica formula (20). We conclude that the logarithmic negativity is exponentially suppressed with the distance $d$ between the subsystems, irrespective of the underlying graph. It is possible to derive a formula similar to (30) for RVB states in the case of adjacent intervals, but we leave this issue to future investigations.

Let us now discuss the physical implications of (71). We consider two regions $A_1, A_2$ of characteristic length $L$ separated by a distance $d$. For continuum theories, such as a massive scalar or conformal field theories (CFTs), the logarithmic negativity is a scaling function of ratios constructed from the characteristic length scales of the system. For gapped theories with a finite correlation length $\xi$, one expects the logarithmic negativity to vanish exponentially for $d/\xi \gg 1$, whereas for CFTs it is a scaling function of the ratio $d/L$ and decays for large values thereof [88,89]. Expression (71) implies that for RVB states the logarithmic negativity vanishes exactly in the scaling limit $d, L \to \infty$ with fixed ratio $d/L$, even for arbitrarily small values of $d/L$. Moreover, our results hold irrespective of the underlying graph. For critical RVB states defined on bipartite graphs, while the correlation functions of certain observables exhibit a power-law decay, entanglement between disjoint regions is nonetheless suppressed exponentially fast in $d$. This is in stark contrast with the CFT behavior. The case of gapped RVB states is also surprising, since the exponential decay of the logarithmic negativity is independent on the correlation length, and is faster than for generic gapped theories. The scaling behavior of the logarithmic negativity (71) is thus highly nongeneric.

There is a substantial difference between the logarithmic negativity and mutual information of disconnected subsystems in RVB states, similarly as for RK states (see Sec. 2.5.2). The mutual information serves as an upper bound for correlation functions [99], and therefore it decays as a power-law for critical RVB states. For gapped RVB states, the decay of the mutual information depends on the ratio $d/\xi$. In both cases, the mutual information is much larger than the logarithmic negativity.

## 3.7 Generalization to SU($\mathcal{N}$) RVB states

We discuss the generalization of our results for SU(2) RVB states to SU($\mathcal{N}$), where spins have $\mathcal{N} = 2S + 1$ components. The idea of SU($\mathcal{N}$) RVB states originates from [96], where the authors investigate SU($\mathcal{N}$) Heisenberg models using Monte Carlo algorithms. We consider a spin-$S$ generalization of the SU(2) singlet state between sites $x$ and $y$, defined as

$$|S_{x,y}\rangle = \frac{1}{\sqrt{2S+1}} \sum_{m \in \{-S, -S+1, \ldots, S\}} (-1)^{m-S} |m\rangle_x \otimes |-m\rangle_y, \tag{72}$$

where $|m\rangle$ is an eigenvector of the magnetization operator $S^z$, with eigenvalue $m$. For $\mathcal{N} = 2$ (i.e. $S = 1/2$), we recover the SU(2) spin singlet of (35), whereas for $\mathcal{N} > 2$, the operator $S^z$ can be constructed from the generators of the SU($\mathcal{N}$) algebra, see [96]. Similarly to the SU(2) case, the SU($\mathcal{N}$) RVB state is an equal-weight superposition of states corresponding to singlet configurations on a graph. Given a singlet configuration $\gamma$, the associated state is

$$|\gamma\rangle = \bigotimes_{(x,y) \in \gamma} |S_{x,y}\rangle, \tag{73}$$

exactly as for SU(2). The difference is that the overlap between states corresponding to different singlet configurations is now [96,97]

$$\langle \gamma | \gamma' \rangle = \mathcal{N}^{n_\ell(\gamma, \gamma') - N/2}, \tag{74}$$

similarly as (36). In the limit $\mathcal{N} \to \infty$, singlet configurations become orthogonal, as for dimer RK states. Indeed, SU($\mathcal{N}$) RVB states interpolate between SU(2) RVB states and dimer states [51].

The calculations of Secs. 3.3 through 3.6 can be generalized to the SU($\mathcal{N}$) case. The reduced density matrix has the form of (49), except that the boundary spins take value in $\sigma \in \{-S, -S+1, \ldots, S\}$, instead of $\sigma \in \{\uparrow, \downarrow\}$. The overlaps that appear in the matrix elements of $\rho_{A_1 \cup A_2}$ can still be interpreted in terms of transition graphs with singlet loops and strings that connect fixed boundary spins. Since singlet states have zero magnetization, the connectivity rules for singlet strings based on the color of boundary spins still holds, but singlet strings of length $n_D$ now have weight $\mathcal{N}^{-n_D/2}$. The reduced density matrix is thus separable, up to terms of order $\mathcal{N}^{-d/2}$, and the logarithmic negativity satisfies

$$\mathcal{E}(A_1 : A_2) = \mathcal{O}(\mathcal{N}^{-d/2}). \tag{75}$$

In the limit $\mathcal{N} \to \infty$, we recover our results for the dimer states, namely we find that the reduced density matrix is exactly separable and the logarithmic negativity vanishes identically for disjoint subsystems.

## 4 Multipartite separability

Thus far, we have focused on the separability of bipartite mixed states constructed from tripartite *pure* states by considering their reduced density matrix on two disconnected subsystems. In this section, we investigate multipartite separability of RK and RVB states.

A system $A$ with $k$ parties, $A = \bigcup_{j=1}^{k} A_j$, in a general state is $k$-separable if its reduced density matrix can be written as

$$\rho_{\bigcup_{j=1}^{k} A_j} = \sum_{i_1, \ldots, i_k} p_{i_1 \ldots i_k} \bigotimes_{j=1}^{k} \rho_{A_j}^{(i_j)}, \tag{76}$$

where $p_{i_1 \ldots i_k}$ are probabilities that sum to one, and $\rho_{A_j}^{(i_j)}$ are Hermitian positive semidefinite operators, as in (1).

### 4.1 RK states

We consider RK states defined on an arbitrary lattice which is divided in $k+1$ subregions, $A_1, \ldots, A_k$ and $B$. The $A_j$'s are disjoints and share a boundary with $B$. The respective boundary configurations are denoted $i_j$. Using similar conventions as in Sec. 2.2, we decompose the state corresponding to a configuration $c$ as

$$|c\rangle = |b\rangle \bigotimes_{j=1}^{k} |a_j, i_j\rangle, \tag{77}$$

the locality of the energy functional $E(c)$ yields

$$E(c) = E(b, i_1, \ldots, i_k) + \sum_{j=1}^{k} E(a_j, i_j), \tag{78}$$

and the RK wavefunction (3) reads

$$|\psi\rangle = \sum_{i_1, \ldots, i_k} \left( \prod_{j=1}^{k} \mathcal{Z}_{A_j}^{i_j} \right)^{1/2} \left( \frac{\mathcal{Z}_B^{i_1 \ldots i_k}}{\mathcal{Z}} \right)^{1/2} |\psi_B^{i_1 \ldots i_k}\rangle \bigotimes_{j=1}^{k} |\psi_{A_j}^{i_j}\rangle, \tag{79}$$

where the subsystems RK states and partition functions are defined as in (6).

Using similar techniques an in Sec. 2.4, we investigate the $k$-separability of the RK state (79). For dimer RK states, the reduced density matrix corresponding to $k$ disjoint regions reads

$$\rho_{\bigcup_{j=1}^{k} A_j} = \sum_{i_1,\ldots,i_k} \left( \prod_{j=1}^{k} \mathcal{Z}_{A_j}^{i_j} \right) \frac{\mathcal{Z}_B^{i_1\ldots i_k}}{\mathcal{Z}} \bigotimes_{j=1}^{k} \rho_{A_j}^{(i_j)}, \tag{80a}$$

where the density matrices for each subsystem are

$$\rho_{A_j}^{(i_j)} = \frac{1}{2} \sum_{i'_j \sim i_j} \sqrt{\frac{\mathcal{Z}_{A_j}^{i'_j}}{\mathcal{Z}_{A_j}^{i_j}}} \left( |\psi_{A_j}^{i_j}\rangle\langle\psi_{A_j}^{i'_j}| + |\psi_{A_j}^{i'_j}\rangle\langle\psi_{A_j}^{i_j}| \right). \tag{80b}$$

This state is exactly $k$-separable, see (76).

For generic RK states, the arguments of Sec. 2.4.2 carry through to the multipartite situation and we find that the state is separable in the thermodynamic limit where the boundary energies are negligible compared to the bulk energy of system $B$.

## 4.2 RVB states

Let us now consider an SU(2) RVB state on an arbitrary graph with $k+1$ subregions, $A_1,\ldots,A_k$ and $B$. The graph distance between two subsystems $A_i$ and $A_j$ is $d_{ij} > 0$, and we define

$$\begin{aligned} d_{\min}^{(i)} &\equiv \min_{\substack{j=1,\ldots,k \\ j\neq i}} \{d_{ij}\}, \\ d_{\min} &\equiv \min_{\substack{i,j=1,\ldots,k \\ i\neq j}} \{d_{ij}\}. \end{aligned} \tag{81}$$

As in Sec. 3.2, we denote by $e_i$ the boundary singlet configuration between $A_i$ and $B$. The reduced density matrix of subsystem $A = \bigcup_{j=1}^{k} A_j$ takes the form (49) generalized to $k$ boundaries. The function $\mathcal{F}$ (see Sec. 3.3) now depends on $2k$ boundary singlet configurations, $\mathcal{F} \equiv \mathcal{F}(e_1,\ldots,e_k;e'_1,\ldots,e'_k)$. Using similar graphical arguments as in Sec. 3.5, it can be shown that terms that break the symmetry $e_i \leftrightarrow e'_i$ in $\mathcal{F}$ correspond to transition graphs where at least one string connects $A_i$ to another subregion $A_j$. Then proceeding as in Sec. 3.5, we find

$$\mathcal{F}(e_i;e'_i) = \mathcal{F}(e'_i;e_i) + \mathcal{O}(2^{-d_{\min}^{(i)}/2}), \tag{82}$$

and hence

$$\mathcal{F} = \mathcal{F}^s + \mathcal{O}(2^{-d_{\min}/2}), \tag{83}$$

where $\mathcal{F}^s$ is the part of $\mathcal{F}$ which is fully symmetric under all exchanges $e_i \leftrightarrow e'_i$. Following Sec. 3.5.2, we conclude that the RVB reduced density matrix of $k$ disjoint subsystems is $k$-separable up to terms of order $2^{-d_{\min}/2}$. In particular, we recoved the exact separability in the scaling limit of large system sizes and distances with fixed ratios. Moreover, our results readily generalize to the case of SU($\mathcal{N}$), where the $k$-separability is spoiled only by terms of order $\mathcal{N}^{-d_{\min}/2}$. In the limit $\mathcal{N} \to \infty$, we recover the exact $k$-separability of the dimer RK states, similarly as in Sec. 3.7.

## 5 Discussion

We have investigated entanglement and separability of RK and RVB states. The first part of this work was devoted to RK states constructed from the Boltzmann weights of an underlying

classical model. We proved the exact separability of the reduced density matrix of two disconnected subsystems for dimer RK states on arbitrary (tileable) graphs, implying the absence of entanglement between the two subsystems. For more general RK states with local constraints, we showed that the reduced density matrix of two disjoint subsystems is exactly separable on the square lattice when the boundaries do not have concave angles. For arbitrary graphs or boundaries with concave angles, we argued that the reduced density matrix of disjoint systems is separable in the thermodynamic limit. We also showed that any local RK state has zero negativity for disjoint subsystems, even if the density matrix is not exactly separable. Such RK states are thus bound states whose entanglement cannot be distilled.

For adjacent subsystems, we derived an exact formula for the logarithmic negativity of RK states in terms of partition functions of the underlying statistical model. Finally, we verified that our results reduce to the Rényi entropy $S_{1/2}$ for complementary subsystems, and argued that the logarithmic negativity satisfies an area law.

Similarly to dimer RK states, RVB states are constructed from a classical dimer model on an arbitrary tileable graph, although the degrees of freedom are spins located on the sites of the graph rather than on the edges. For spin $1/2$, we showed that the reduced density matrix of disconnected subsystems is separable up to exponentially small terms of order $2^{-d/2}$, where $d$ is the lattice distance between the two subsystems. Separability thus holds in the scaling limit, even for arbitrarily small ratio $d/L$, where $L$ is the characteristic size of the subsystems. While asymptotic separability and vanishing logarithmic negativity in the limit of large separation is a usual feature of local theories, the fact that they hold in the scaling limit with arbitrarily small ratio $d/L$ is a novel, surprising feature of RVB states.

For simplicity, we mainly focused on SU(2) RVB states (i.e. with spin $S = 1/2$), but our results straightforwardly generalize to SU($\mathcal{N}$). In particular, we argued that separability for two disjoint subsystems holds up to exponentially small terms of order $\mathcal{N}^{-d/2}$ and that the logarithmic negativity is exponentially suppressed as $\mathcal{O}(\mathcal{N}^{-d/2})$ with the distance $d$ between the subsystems, irrespective of the underlying lattice. Finally, in the limit $\mathcal{N} \to \infty$, we recover the results of dimer RK states, namely the reduced density matrix of disjoint subsystems is exactly separable, and the logarithmic negativity vanishes.

We extended our analysis to the multipartite situation, considering the separability properties of $k$ disconnected subsystems. Similarly as in the bipartite scenario, we found that the reduced density matrix is exactly $k$-separable for the dimer RK states, whereas separability is spoiled only by subleading terms that vanish in the scaling limit for generic RK states and RVB states. Hence, for disjoint subsystems, there is neither bipartite nor multipartite entanglement in these states in the scaling limit, irrespective of the underlying lattice.

We conclude with an outlook on future directions. First, RK states are examples of *sign-free* states since they are defined as a coherent superposition of basis states with positive coefficients. Sign-free states are groundstates of stoquastic local Hamiltonians (see, e.g., [100, 101]). For one-dimensional systems with zero correlation length, the *measurement-induced entanglement* (MIE) [102] of such non-negative states is superpolynomially small in the distance between two subsystems [103, 104], which was conjectured to hold as well in higher dimensions. The MIE is the amount of entanglement that can be generated between two subsystems if one measures the rest of the system; it can thus be regarded as a measure of entanglement between noncomplementary subsystems. Our results suggest that the logarithmic negativity of RK and RVB states is smaller than the MIE. It would be worth investigating the relation between these two entanglement measures in the context of sign-free states. Second, our results for RK states on graphs are consistent with the literature regarding the separability of the reduced density matrix for continuum RK states, see [77]. It would be interesting to see whether such a continuum treatment is amenable in the context of field theories describing spin liquids. Third, one could generalize our results on the logarithmic negativity of adjacent

subsystems for RK and RVB states to arbitrary graphs and partitions, pushing toward a more quantitative understanding of its behavior.

# Acknowledgements

We thank Jean-Marie Stéphan, Christian Boudreault and Bryan Debin for interesting discussions and comments on the manuscript. We also thank Antoine Brillant for previous related work.

**Funding information**    G.P. holds a CRM-ISM Postdoctoral Fellowship and acknowledges support from the Mathematical Physics Laboratory of the CRM. C.B. was supported by a CRM-Simons Postdoctoral Fellowship at the Université de Montréal. W.W.-K. was funded by a Discovery Grant from NSERC, a Canada Research Chair, and a grant from the Fondation Courtois.

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
