# Peer review of "Separability and entanglement of resonating valence-bond states"

_SciPost Physics, doi:SciPost Phys. 15, 066 (2023)_

## Round 1 · Referee Report · Anonymous (Referee 1) · 2023-4-3

Strengths

1-The Authors provide a clear and systematic investigation of the separability and entanglement of RK and RVB states, a class of states constructed from an underlying classical model on an arbitrary graph, that are of great interest in condensed matter physics.

2-The paper is extremely well written, and all the passages and calculations are presented in a very clear way, so that the reader can easily follow them.

3-The properties of separability and entanglement, derived explicitly for some specific examples, are also found to be valid in more general cases, as they hold for arbitrary underlying lattices.

Weaknesses

1-There are no particular points of weakness.

Report

In this paper, the Authors study the separability and entanglement of the Rokhsar-Kievelson (RK) and resonating valence bond (RVB) states. To determine the separability of such states, they provide a systematic derivation of the corresponding reduced density matrices (RDM) for systems consisting of several disjoint intervals. In the case of systems consisting of two disjoint and adjacent intervals, they also evaluate the logarithmic negativity.
In the first part, the Authors study the separability of the RDM of disjoint subsystems for RK states. They show that the RDM is separable for any RK state with local constraints if the underlying graph is a square lattice and the two disjoint subsystems have no concave angles. They also show that the RDM is separable for arbitrary underlying graphs in the case of dimer states, that are RK states whose underlying statistical model is the dimer model. Hence, they prove that in these cases the two disconnected regions are not entangled. In the more general case of RK states with local constraints, they find that the RDM is separable in the thermodynamic limit. In this case, they also find that the logarithmic negativity vanishes, even if the RDM is not exactly separable.
In the second part, they repeat the same analysis for RVB states, where the degrees of freedom are represented by spins located on the sites of the graphs, showing that the RDM of disconnected subsystems is separable up to exponentially small terms in the distance between the subsystems. Moreover, the logarithmic negativity decay exponentially with such distance. These results hold irrespectively of the underlying graphs.

The paper provides a detailed characterization of the entanglement properties of RK and RVB states, and opens to many possible future investigations. For these reasons, I recommend publication on Scipost.

---

## Round 1 · Referee Report · Anonymous (Referee 2) · 2023-4-4

Strengths

1-The paper shows the separability of the reduced density matrix of two disconnected systems and zero value of logarithmic negativity in Rokhsar-Kivelson states and resonating valence bond states by exact computations on the lattice.
2-The methods apply to arbitrary lattices so the results are very general.

Weaknesses

1-It is suggested that the results go beyond lattices to tileable graphs, but this point is not particularly clarified and no example is given. 2- The boundary conditions seem to be totally disregarded in the paper, even though they are are known to be important in tiling problems and can influence even the bulk free energy (see e.g. J. Phys. A 33 No. 40 (2000), 7053). For example, in eq. (33) in the paper the authors assume, without any comment, that the fixed dimer configurations on the boundaries only affect the subleading coefficients. Why is it expected that the boundary conditions can be neglected in this problem?

Report

The paper deals with the investigation of separability of the reduced density matrix and the computation of logarithmic negativity in two classes of states that are of interest in condensed matter physics, Rokhsar-Kivelson states and resonating valence-bond states. The results are obtained on the lattice by analytical methods and are in agreement with the existing field theoretical results, mentioned in the paper. The reduced density matrix is computed by labeling the boundary configurations, that live on the edges of the subsystems, counting the configurations corresponding to each boundary configuration and computing the relevant overlaps. The emphasis is on disconnected subsystems, for which the paper obtains a zero value of logarithmic negativity and a stronger result of the separability of the reduced density matrix. As a side product, the results can be used to conclude that nonvanishing value of mutual information in such systems is not due to quantum entanglement.

In my opinion, the paper is well written and the results and the methods are interesting enough for publication in SciPost Physics. But before recommending I would like that the authors address the following points.

Requested changes

1-Address the above-mentioned weaknesses.
2-In the caption of Figure 1 it might be good to indicate that regions $A_1$ and $A_2$ are defined by dotted lines. The current version might suggest that $A_1$ is everything covered by green dimers.
3-It is written that "the logarithmic negativity is exponentially suppressed with distance $d$ between the subsystems, irrespective of the underlying graph." Similar phrases for exponential suppression are used elsewhere in the paper. Is there a bound for negativity that is independent of the choice of $A_1$,$ A_2$ or there is no such uniform bound?
4-In eq. (4) and afterwards the ket $|a_k,i>$ is defined with respect to some bulk configuration of $A_k$ with fixed boundary configuration $i$. Correspondingly, it is a vector that belongs to a vector space of dimension that depends on the difference of the size of $A_k$ and the number of boundary dimers. Please clarify whether in the braket $<a_k,\ell | a_k',\ell'>$ the two vectors can have different dimensions or it is a true scalar product. Similarly in eq. (8).
5-The last sentence of the abstract is written as if the results cover all known gapless quantum critical systems . Please clarify.

  • validity: high
  • significance: high
  • originality: good
  • clarity: good
  • formatting: excellent
  • grammar: perfect

Author:  Gilles Parez  on 2023-04-19  [id 3601]

(in reply to Report 2 on 2023-04-04)

Dear Referee,

Thank you for your thorough analysis of our paper, your positive review and the various points you raised. We answer them below, using the same numbering as in your review.

Gilles Parez, for the authors

Weaknesses:

  1. Indeed, our results hold for arbitrary graphs, unless stated otherwise (for example Secs. II.C.2 and II.E.3 pertain to the square lattice). Apart from these examples, all of our calculations are performed irrespective of the underlying lattice, and hence by construction apply to any graph. For the sake of clarity, we only draw simple lattices in the figures, such as the square and triangular lattices, but all the calculations are general. Hence, we claim that the fact that our results hold for arbitrary lattices is not suggested, but indeed transparent, by construction.

  2. The only place where the boundary conditions could play a role is in the discussion around Eq. (33), as you rightfully point out. Indeed, there are situations in which the boundary conditions of the dimer model can have a strong impact on the allowed configurations, even deep in the bulk, and hence affect the bulk free energy. These situations are often related to so-called arctic phenomena (for example, the six-vertex model with domain wall boundary conditions, studied in the reference you mentioned, is related to the dimer model on the Aztec diamond, where arctic curves are known to appear). In our discussion around Eq. (33), we consider rectangular regions of the square lattice, with fixed boundary dimers. To the best of our knowledge, there is no arctic phenomenon in that case, and hence we expect boundary configurations to only affect subleading terms. Furthermore, if we had boundary-dependent bulk free energies, this could lead to a volume-law for the logarithmic negativity, which is physically forbidden for local theories, such as those we consider.

Requested Changes:

  1. See above.

  2. We added a sentence in the caption of Fig. 1 to clarify the definitions of $A_1$ and $A_2$.

  3. We are not aware of such a bound. This is a very good question, which we leave open for further investigations.

  4. In $|a_1,i\rangle$ (it is similar for $a_2,j$) it is true that $|a_1\rangle$ is defined in a Hilbert space which corresponds to all the edges in $A_1$, from which we removed the edges occupied in the boundary configuration $i$. Thus, the ket $|a_1, i\rangle = |a_1\rangle \otimes |i\rangle$ is always well defined in the Hilbert space corresponding to the whole system $A_1$, irrespective of the boundary $i$. Therefore, the product $\langle a_1,i|a’_1,i’\rangle$ is a scalar for all $i,i’$. For the product in Eq. (8) there is no ambiguity either. Indeed, in the definition of $|\psi^{i,j}_B\rangle$, see Eq. (6b), the state is a linear superposition of vectors $|b\rangle$ which are defined in the Hilbert space corresponding to the edges in B, irrespective of the boundaries $i,j$. Thus, the product in (8) is a scalar for all $i,i’,j,j’$.

  5. We added the word “certain” in the last sentence of the abstract, to highlight that, indeed, RK and RVB states do not encompass all possible critical and gapped states.

---

## Round 2 · Referee Report · Anonymous (Referee 2) · 2023-5-26

Report

The authors have addressed well the points raised in my report. I recommend the paper for publication.

---

## Round 2 · List of Changes

1) We added a sentence in the caption of Fig. 1 to clarify the definitions of $A_1$ and $A_2$. 2) We added the word “certain” in the last sentence of the abstract.

---

## Editorial Decision

published